# Self-Healing Materials for Electronics Applications

**DOI:** 10.3390/ijms23020622

**Published:** 2022-01-06

**Authors:** Fouzia Mashkoor, Sun Jin Lee, Hoon Yi, Seung Man Noh, Changyoon Jeong

**Affiliations:** 1School of Mechanical Engineering, Yeungnam University, Gyeongsan 38541, Korea; mkfzia@gmail.com; 2Research Center for Green Fine Chemicals, Korea Research Institute of Chemical Technology, Ulsan 44412, Korea; sunee6210@gmail.com; 3Mechanical Technology Group, Global Manufacturing Center, Samsung Electro-Mechanics, 150 Maeyeong-ro, Yeongtong-gu, Suwon 16674, Korea; Poemisty@gmail.com

**Keywords:** self-healing materials, extrinsic self-healing materials, intrinsic self-healing materials, energy storage devices, bioelectronic devices

## Abstract

Self-healing materials have been attracting the attention of the scientists over the past few decades because of their effectiveness in detecting damage and their autonomic healing response. Self-healing materials are an evolving and intriguing field of study that could lead to a substantial increase in the lifespan of materials, improve the reliability of materials, increase product safety, and lower product replacement costs. Within the past few years, various autonomic and non-autonomic self-healing systems have been developed using various approaches for a variety of applications. The inclusion of appropriate functionalities into these materials by various chemistries has enhanced their repair mechanisms activated by crack formation. This review article summarizes various self-healing techniques that are currently being explored and the associated chemistries that are involved in the preparation of self-healing composite materials. This paper further surveys the electronic applications of self-healing materials in the fields of energy harvesting devices, energy storage devices, and sensors. We expect this article to provide the reader with a far deeper understanding of self-healing materials and their healing mechanisms in various electronics applications.

## 1. Introduction

Synthetic materials have been extensively applied in a diversity of fields; however, these materials are prone to deterioration and failure [1]. Small crevices, voids, and microcracks in the materials can cause early damage if they are not identified or repaired in time [2]. Some of these damages occur at the microscopic level within the structure of the material, which makes them difficult to repair manually and results in damage to the products before the end of their operational life [3]. By contrast, living organisms can repair their injuries and automatically recommence their original functions, which extends the longevity and survivability of most living organisms [4]. Such a potent biological function has inspired researchers to develop a novel class of synthetic materials that are known as self-healing materials (SHMs) [5]. This concept has become an important area of research interest and has technological importance in prolonging the lifespan of synthetic materials. SHMs are a class of smart materials with the propensity to autonomically and spontaneously repair themselves and recover their functionality after being damaged [3,6,7,8]. With such materials, one can envisage extremely durable and reliable materials with low maintenance costs. The growth of this field has unlocked an era of new smart materials.

A wide range of SHMs have already been developed. Although these materials have different structures and compositions, they all use simple chemical reactions as part of their healing mechanism. On the basis of the chemistry that is involved, the mechanisms by which SHMs function can be expeditiously classified into two conceptually different groups: autonomic and non-autonomic self-healing (SH) approaches [9]. Autonomic SH mechanisms proceed with the aid of healing agents that are automatically released immediately after the damage occurs and facilitate the repair. However, non-autonomic SH mechanisms proceed via a reversible or dynamic chemical bond formation and are triggered by applying a stimulus (e.g., light or heat). The former mechanisms do not entail human interference whereas the latter mechanisms do [9,10]. SHMs can also be conceptually categorized into intrinsic and extrinsic SHMs [3,10], as will be discussed in detail in Section 2.

The potential for decreasing the chance of failure and improving the product life by incorporating SHMs into materials has attracted great attention from scientists. Every year, a remarkable number of works that are related to the development of SHMs and their practical applications are reported. Advances in SHMs in the area of organic electronic materials have been well demonstrated and have garnered intensive attention. Self-healable organic electronic materials have been used in devices such as energy-harvesting devices, field-effect transistors, sensors, and electronic skin [11,12,13,14,15,16]. Over the past few years, important advances in techniques for constructing organic electronics with high performance and high safety have been achieved. These advances were made possible by the development of an extensive range of compatible substrates with good mechanical flexibility and by reductions in the cost of materials and processes, which have, in turn, moved organic electronics toward commercial viability [17]. Nevertheless, the durability and stability of the materials remain a challenge. Hence, the application of the SHM concept to organic electronic devices offers a new approach to the development of safer, sustainable materials with a long working life [18]. Although technical difficulties are encountered in maintaining the electronic performance of SHMs during the SH process, numerous advances in the design and production of SH polymeric materials, metals, conductors, ceramics, and concrete materials have nonetheless been achieved [19,20,21,22,23]. In this article, our goal is to provide an up-to-date overview of SH organic electronic materials for numerous applications. In the subsequent sections, we provide a general overview of the SH concept and briefly outline the different SH mechanisms as well as the associated critical concerns with these approaches.

## 2. Self-Healing Mechanisms

Self-healable polymeric materials are among the most extensively researched materials. The enormous freedom of design of polymeric systems enables the application of a myriad of innovative methods to facilitate healing. Self-healable polymeric materials are usually categorized into two different groups—extrinsic and intrinsic—on the basis of the approach that is used to incorporate healing functionality into the bulk material. These groups are further subcategorized, as detailed in the following subsections.

### 2.1. Extrinsic Self-Healing Materials

Extrinsic SHMs themselves do not have self-repairing ability; the healing process depends on an external agent [24]. A healing agent that is encapsulated in microcapsules or microvascular networks must be enclosed within the materials during their production [1]. After or at the onset of fracture in the materials, the capsules or vascular networks rupture, releasing the compartmentalized healing agent via capillary action; the healing agent then seals the damaged surface and prevents propagation of the damage [25]. Materials that are employed as a liquid healing agent have particular features: they are flowable (to seal the fracture effortlessly by capillary action), highly stable (to enable long-term storage in the vascular network or capsule without reacting), and rapidly react with the matrix (for swift mending) [26,27]. The mechanical properties of the healing agent should be similar to those of the material into which it is inserted [27,28].

#### 2.1.1. Microcapsule Embedment

Encapsulation methods are mostly applied to polymers and coatings. Material healing is achieved by inserting compounds with reactive groups or healing functionality into capsules, followed by chemical reactions [24,29]. Various processes such as ring-opening metathesis polymerization, cross-linking reactions, cycloaddition, cycloreversion, or mechanochemical catalytic activation are involved in these chemical reactions [24]. Fracture in the material functions as a stimulus to activate the repair procedure. A crack leads to breaking of the microcapsules, allowing the healing agent to flow and reach the fracture location by capillary action, where it spreads over the two rupture surfaces because of the surface tension [30]. In addition, precursors interact with the adjacent inserted catalysts to form a network that stops the further growth of the fracture and restores the mechanical integrity by continuing the aforementioned reactions (Figure 1) [29]. The first structural polymeric material with a microencapsulated healing agent with the ability to autonomously heal damage was reported by White et al. [31]. They reported ~75% crack recovery. Damage in the material that they developed caused the release of dicyclopentadiene (DCPD) monomer that was enclosed within microcapsules into the fracture area via capillary action. A ring-opening reaction was activated when the DCPD comes in contact with the inserted Ru-based Grubbs catalyst, resulting in repair of the damage.

The majority of the reported microcapsules for capsule-based SH studies were in the range of 100–200 mm at 10–20 wt% catalyst loading [31,32,33,34]. This is to ensure that adequate healing agents are available to fill the fracture volume to attain excellent healing efficiency [35]. However, when we come across its possible applications such as adhesives and thin coatings, a smaller scale of capsule size is preferred. Blaiszik et al. [36] designed nanocapsules with an average size of 220 nm, whereas Kirkby et al. [37] implemented shape-memory alloy (SMA) wire into the composite to reduce the crack volume and increasing the crack fill factor.

The most difficult part of the encapsulation strategy is controlling the release of the precursor. Alternative sustainable catalyst-free repair processes are also being explored because the catalyst-based strategy is not economical and requires some sacrifices with respect to the mechanical properties of the material. The performance of a given encapsulation approach is determined by a number of critical factors [38,39]. Encapsulation methods for healing are carried out by one of four methods: (i) phase separation of either the repairing agent or the catalyst in the matrix material, (ii) combining the encapsulated liquid agent with a dispersed catalyst, (iii) placing the catalyst and repairing agent in different capsules, or (iv) direct reaction of the healing agent with the matrix’s functionality under the influence of an external stimulus [27]. Nonetheless, the inherent shortcoming of the capsule healing technique is that the healing agent is no longer available in the same region after a single cracking event occurs [10].

#### 2.1.2. Microvascular Embedment

A microvascular technique is inspired from the respiratory system of living organisms. The composite materials that have been explored for self-repairing purposes are usually composed of very fine hollow fibers and a mesoporous-structured compound and can effectively improve the life of the structural material into which they are incorporated. The incorporation of micro-channels within a polymeric composite allows multiple healing [24]. The operating principle of the vascular healing system is analogous to that of the microcapsule technique, although the fabrication and matrix integration methods differ between the two techniques. In the case of microvascular embedment, the healing agent is incorporated into a complex network of hollow vessels or canals and this interconnected network is retained until the fractures are repaired. To improve the SH capability of a vascular system, achieving a better interaction between the healing agents, polymeric matrix substrate, and the network materials is imperative [40,41]. The vascular networking system can have different dimensionalities (e.g., a one-dimensional (1D), two-dimensional (2D), or three-dimensional (3D) network) (Figure 2a) [42,43]. Unlike the capsule-based technique, the vascular technique involves introducing healing agents after the network has been assimilated into the matrix. The surface wettability, viscosity, and chemical reactivity of the substance all play roles in the selection of a healing agent. High viscosity and poor wetting performance prevent the efficient filling of the network, and chemical incompatibility jeopardizes the system’s long-term stability. These characteristics affect the design of the vascular system—mostly the diameter and size of the chosen vessel—because its wetting and viscous behavior affect the discharge and diffusion of the repairing agent [44].

The mechanical properties of a matrix with an embedded network are influenced by the volume fraction, stiffness of the wall, channel distribution, uniformity, and the association between the matrix and the network. Similar to the development of the capsule-based method, the activating mechanisms of the microvascular approach have been validated and the repairing capability has been characterized. Importantly, the vascular technique enables recurrent repairing of successive fracture events because of access to a vast reservoir of healing agent and the ability to replenish the network (Figure 2b) [45].

Toohey et al. created a bioinspired microvascular network coating/substrate design [46]. In their 3D microvascular network design, the healing agent is continuously supplied to the fracture area. Damage to an epoxy coating was repaired repeatedly. However, diminution of the integrated catalyst and the necessity to re-supply multiple healing agents within these designs constrain their repairing ability. To overcome this restriction, Toohey et al. [47] photolithographically patterned four isolated areas within the embedded microvascular network. This improved scheme not only enables the development of new healing chemistries but ensures that neither healing agent is depleted. In this research, as many as 16 recurrent repairing events were achieved out of 23 cycles. A main shortcoming of this method is that the healing agents must transfer and diffusively mix over long distances within the damaged area, which corresponds to approximately one-half the thickness of a given isolated microvascular region. Williams et al. subsequently reported a self-healable sandwich composite that comprises of either single [48] or dual [49] fluidic networks.

#### 2.1.3. Hollow-Fiber Embedment

The healing mechanism of hollow fiber SHM involves the insertion of hollow fiber into the matrix material. The hollow fiber is equipped with a healing agent so that when the material is cracked or injured, the repair fluid from the hollow fiber is released and bonds the crack (Figure 3). When the composite material is injured, the crack propagation force causes the liquid core fiber to burst and releases the healing agent to heal the damage and restore the mechanical properties of the material. Hollow glass fibers as big as 30 μm^2^ and as small as 5 μm^2^ [32,50] were developed. Though a low mending efficiency was presented for smaller diameter hollow glass fibers and higher mending efficiency of 93% was attained for the larger diameter hollow glass fibers under impact test, fiber blockage was still present in both cases. This is owing to the low volume filling into such tiny fibers as well as rapid curing of one-part epoxy resin system [32].

Belay et al. [51] used smaller hollow glass fibers (Hollex fibers) (external diameter of 15 μm, internal diameter 5 μm) that were loaded with resins. A composite system that was based on these filled glass fibers could not transport the resin into the damaged area because highly viscous epoxy resins were used and healing performance was also poor. Later, Bond et al. improved the manufacturing of the hollow glass fibers [52] and used these fibers as containers for repairing agents and/or dyes [53,54,55]. The diameters of these borosilicate fibers ranged from 30 to 100 μm and they were reported to have hollowness ratio of ~55%.

These hollow-fiber-based SH composites comprising of hollow fibers can be restored by repairing agents which can recover ~97% of the composites’ basic flexural strength [56]. The benefits of such a self-repairing material include a large volume of accessible healing agent to repair fractures, the feasibility of using numerous activation methods and types of resins, and the ability to visually inspect the broken area. However, the limitations of this method are that the fracture of hollow fibers is mandatory to discharge the repairing agent, low-viscosity of healing agent is favorable to facilitate fiber infiltration, and, in addition, the use of hollow glass fibers in carbon-fiber-reinforced plastics influences the coefficient of thermal expansion, which introduces another problem [56,57].

### 2.2. Intrinsic Self-Healing Materials

The intrinsic SH approach is based on the intrinsic function of a polymeric material and offers the prospect of repair via reversible chemical bonds that enable damage closure and sealing in the materials under a certain form of stimulation (typically heat) [2,58]. In this approach, the healing mechanism depends on the creation of reversible chemical interactions between polymeric chains comprising of dynamic covalent [59,60,61] and noncovalent bonds [7,62,63]. When the material is damaged, weak chemical linkages are disrupted and, because of the dynamic nature of polymeric chains, these broken bonds can re-form at the ruptured interfaces [18]. Reversible noncovalent interactions comprise of hydrogen bonding (H-bonding), metal coordination bonding, electrostatic cross-linking, and others as a main healing mechanism of intrinsic SHMs as shown in Table 1. Compared with the formation of covalent bonds, the formation of these noncovalent bonds does not involve large amounts of energy [64]. Moreover, the assemblies that are formed by these noncovalent bonds have low kinetic stability and more readily undergo reversible intermolecular dissociation and generation [64,65]. By contrast, various reversible covalent bonding reactions, such as cycloaddition and chain exchange reactions, are viable for the formation of such intrinsic SHMs [1]. Reversible covalent bonds in a self-healable polymer enable structural changes and bond arrangement via reversible reactions that are triggered by an external stimulus such as pH, heat, or light. When the external stimulus is removed, the polymers regain their bonding strength and stability, which are comparable to those of the corresponding irreversible covalent polymers [66]. Thus, reversible covalent bonds have strong potential to heal polymeric materials several times, resulting in materials with excellent mechanical toughness [4]. However, because of the high activation energy of the covalent bonds, an enormous amount of energy (light or heat) is always necessary to activate the healing process, which is a major shortcoming [4,65]. Because of this required energy input, intrinsic SHMs are also known as nonautonomous SHMs [18].

#### 2.2.1. Reversible Noncovalent Bonds

##### Ionic Interaction

Ionomeric interactions, which are another method of fabricating SHMs, generally occur when ionomers form in the polymeric structure. Ionomers contain copolymers with a maximum ionic group content as high as 15 mol%; these polymers can form clusters and behave as physical crosslinking points. They enable the reversible formation and re-formation of the network structure [27,56,67]. Mending of a material, which occurs after strong impact, is quick and autonomous and does not require healing agents or an external stimulus (e.g., heat) [1,24]. For a self-healing process to proceed in ionic interaction, the total energy of the process, $\Delta G\left(\mathrm{Gibbs}\;\mathrm{free}\;\mathrm{energy}\right)\;=\Delta H\left(\mathrm{Enthalpy}\right)\;-\;T\left(\mathrm{Kelvin}\;\mathrm{temperature}\right)\Delta S\left(\mathrm{Entropy}\right)\;<0$ for spontaneous condition. Therefore, if enthalpy is decreasing and entropy is increasing during a healing process, then Gibbs free energy must be negative for spontaneous process [68,69]. After this thermodynamic condition is met, the molten polymer surfaces fuse together via inter-diffusion to seal damaged area, resulting in a rearrangement of the ionic clustered region and long-term relaxation for the healing process. In view of this point, the majority of self-healing processes occur at an elevated temperature (melting temperature) or it needs a low glass transition temperature ($T_{g}$) [70,71]. The construction of an SH system that is based on ionomers involves several bonding interactions, including ion–ion, ion–dipole, and dipole–dipole bonds [24]. This type of self-repairing method follows a two-step procedure in which the ionomeric network is disrupted by ballistic impact and the energy that is produced by the friction during the breakage is transmitted to the injured area of the polymer, causing localized melting. Subsequently, the molten surface is joined by interdiffusion to heal the damaged area, followed by re-formation of the structure and long-term relaxation [24,72,73]. As the SH process is mainly dependent on thermally-activated chain diffusion, a larger cluster size that restrains the chains will decrease the rate of repair. However, if the clusters can split in response to the stimuli, this method might accelerate the healing process [4]. This advantage was demonstrated using polyelectrolytes that were prepared from poly(acrylic acid)/poly(allylamine hydrochloride) pairs. Reisch et al. used ultracentrifugation and extrusion in the presence of salt to obtain compact polyelectrolyte complexes; the resultant systems exhibited self-healing features [74]. The self-repairing capability of the systems was found to improve with increasing concentration of sodium chloride, implying that the dynamics of the materials involve disrupting the ionic bonding between the polyelectrolyte pairs by the salt, which drives the movement of the polymeric chains [4].

The influence of ionic concentration on poly(ethylene-*co*-methacrylic acid) copolymers and ionomers has revealed the relation between the diffusibility, elasticity, and the SH capability at different temperatures [75]. An increase in the ionic content causes a greater cross-linking density but hinders the mobility of the polymeric structure, resulting in better elasticity and resilience in the melt, which is not observed in the nonionic varieties. The system formed a strong network around the damaged area to achieve the elasticity that is required for healing. The nonionic varieties lost mending capability upon reaching their melting temperature. A large content of ionic species was unfavorable for mending at lower temperatures because of inelasticity and poor mobility. Such relationships between diffusibility, mending temperature, elasticity, and SH capability apply not only to ionomers but also to other SH systems. 

An imidazolium ion-based photocurable monomer was prepared and self-healable photocured materials were synthesized via a green photopolymerization method in only 20–60 s when 2-hydroxyethyl acrylate, 6-(3-(3(2-hydroxyethyl)-1*H*-imidazol-3-ium bromide)propanoyloxy)hexyl acrylate, isobornyl acrylate, and 2-(2-ethoxyethoxy)ethyl acrylate were added as monomers [76]. The SH and mechanical properties of the materials were modified by changing the number of monomers. The optimized and synthesized material exhibited an elongation at break of 205%, tensile strength of 3.1 MPa, and a healing efficiency of 93%, and the repairing temperature was varied from room temperature to 120 °C.

Daemi et al. [77] designed a self-repairing alginate-based polyurethane elastomer that exhibited excellent biodegradability and mechanical properties and was constructed by supramolecular ionic interactions (Figure 4). The ionic interactions imparted the polyurethane elastomer with strong mechanical properties. The elastomer demonstrated rapid and outstanding self-repairing capability, with an elongation at break of 859.5% and a tensile stress of 48.1 MPa. After the mending process, the healing efficiency in the first and third healing cycles was reported to be 87.3% and 70.2%, respectively. The high affinity of cationic groups in polyurethane chains to the carboxylate groups of alginates was found to influence the mechanical and self-repairing properties.

Chen et al. [78,79] designed a novel zwitterionic polyurethane with multi-shape-memory effect and SH properties. The elongation at break and tensile stress of the as-synthesized material were reported to be 87% and 17 MPa, respectively. The recovery of the elongation at break and tensile stress was 90% and 88%, respectively, when the cut-off material was placed for 30 min under 80% relative humidity (RH) conditions and then dried for 2 h at 50 °C. The self-repairing mechanism was attributed to the spontaneous interaction of zwitterions followed by gradual re-entanglement [78].

##### Hydrogen Bonding

H-bonding is one of the most noticeable modes of reversible chemical interaction. H-bonds are stronger than van der Waals bonds but weaker than covalent bonds [6]. H-bonds can be used to synthesize supramolecular polymers with a wide range of mechanical properties, resulting in polymers that range from supramolecular gels to tough rubber-like materials due to H-bonds’ excellent characteristics (e.g., directionality and versatility) [4,80]. Numerous investigations involving the preparation of self-healable polymers by integrating strong and reversible noncovalent H-bonding moieties into a polymeric structure have been reported, and their approach is recognized as an efficient method of producing SHMs [81,82,83].

Meijer et al. first reported ureidopyrimidinone (UPy) groups, which are among the most important groups for forming H-bonds, in 1997 [84]. The supramolecules that are synthesized by UPy units exhibit numerous functionalities, high heat stability, high strength of quadruple H-bonding interactions, robust mechanical strength, and good creep resistance for reversibility [85,86,87]. Song et al. [88] used a biometric synthesis strategy to synthesize mendable polyurethane elastomers through H-bonding interactions (Figure 5a). The molecular structures of elastomers were composed of hierarchical H-bonds that were designed from urethane, urea, and UPy groups (single, double, and quadruple H-bonding). All of the investigated mechanical parameters (i.e., modulus, strength, toughness, and elongation) were improved with an increasing number of UPy units. The optimized sample with UPy units exhibited a high toughness and tensile strength of 345 MJ m^−3^ and 44 MPa, respectively, with a mending efficiency as high as 90% after a 48 h thermal treatment at 100 °C. You et al. [89] developed PSeD-U bioelastomers by integrating UPy units with strong self-complementary quadruple H-bonds into poly(sebacoyl diglyceride) (PSeD). The PSeD-U exhibited a high SH rate at 60 °C, healing within 30 min, and the healed material showed stretching properties that were similar to those of the original material.

A self-healable photochromic polymeric matrix was designed from a biomass-derived elastomer by incorporating multiple H-bonds (UPy groups) and inducing covalent cross-linking [90]. The healing capability and greater extensibility resulted from the soft characteristics and dynamic nature of the elastomer, whereas the reassociation of the damaged H-bonds was caused by the covalent cross-linking. The synthesized elastomer displayed good toughness (42.76 MJ m^−3^) and high extensibility (>2600%). Moreover, the elastomer exhibited good SH properties (full mending of damage), along with a toughness of ~24.1 MJ m^−3^ and excellent mechanical recovery (elongation of 1900%) after mending at 60 °C for 24 h. Wu et al. [91] reported a complementary approach for designing supramolecular elastomers that were based on UPy units with loosely packed spacer units (Figure 5b). The designed elastomers exhibited high extensibility, excellent strength, and high toughness, with 63.7 MJ m^−3^ of dissipated energy. Also, the elastomer was proficient in recovering mechanical breakage and routine damage, with a self-repairing efficacy of ~90%. Han et al. [92] constructed a self-repairing hydrogel by combining chitosan with graphene oxide (GO). In this research, chitosan chains exist in an ordered structure because of the complex intra-/intermolecular H-bonding. When the material was heated, the H-bonding interactions among the chitosan chains were weakened and the mobility of the chitosan chains and GO nanosheets increased. The stretched chains were likely to intermingle with the GO, and the chance of entanglement increased, ultimately forming a network. The outcome of this work was that a hydrogel was well synthesized under ambient conditions. Chen et al. [93] were the first to report an electrically and mechanically self-healable supercapacitor (SC) that was based on H-bonding. The substrates consisted of a supramolecular network with a glass-transition temperature (*T_g_*) below room temperature and hierarchical flower-like TiO_2_ nanostructures. The H-bonds between the matrix and the conductive material were the main driving force for the SH capability, which could be repaired to 85.7% even after the fifth cutting. Leibler et al. [63] reported an H-bonded network that, upon rupture, could restore the H-bonded patterns when the fractured fragments were pushed toward each other at the fracture site, leading to a self-repaired interface. In this work, fatty di- and tri-acids were used in a two-step synthetic pathway comprising of the condensation of acidic moieties with a controlled excess of diethylene triamine, followed by reactions with urea, to produce a mixture of oligomers containing H-bonding groups such as diamido tetraethyltriurea, di(amidoethyl)urea, and amidoethyl imidazolidone. The as-prepared supramolecular assembly resembled a translucent glassy plastic, with a *T_g_* of 28 °C; it behaved like a soft rubber when it was heated above its *T_g_* (to 90 °C) and demonstrated SH capability at room temperature. The proposed mechanism explaining these self-repairing properties lies in the dynamics and density of strongly associating H-bonding groups at the ruptured surface. Upon fracture of the elastomers, the H-bonds of the supramolecular network rather than the covalent bonds were broken, resulting in a high density of non-associated H-bonds at the interface. When the newly fractured fragments contacted the free H-bonding units on the surface of the material, the fragments associated across the fracture. This method enabled the film to be repaired and could restore the material’s original mechanical properties. When the fractured interfaces were out of contact for a long period (more than six hours), the number of non-associated H-bonding units that were available for healing decreased and the self-repairing process was inefficient (only 50% recovery after 6 h apart). Wang et al. [94] observed a similar time-dependent repairing method in a mendable supramolecular hydrogel system. Moreover, a newly damaged surface was found to not adhere to the nonbroken surface of the material at room temperature, indicating that non-associated H-bonds at the surface are a prerequisite for the repairing method.

H-bonding are extensively utilized in the synthesis mechanism of SHM owing to their high activity, easy formation, and thermal reversibility which can also be used as auxiliary bonds to optimize the healing ability and mechanical properties of original SHMs. The bottleneck issues for their practical applications are long repairing time, lower mechanical strength, and aging sensitivity. Presently, a variety of elastomers with an SH ability and reprocessability have been created by dynamic chemistry to prolong the service life, improve the reliability of polymeric materials, and reduce the waste. SH elastomers that are made by applying the H-bond mechanism have the characteristics of rapid rate of healing and good mechanical properties, and are extensively utilized in electronic, mechanical, and biological fields such as electronic skins, robotic intelligent arms, and new sensors [95].

**Figure 5 ijms-23-00622-f005:**
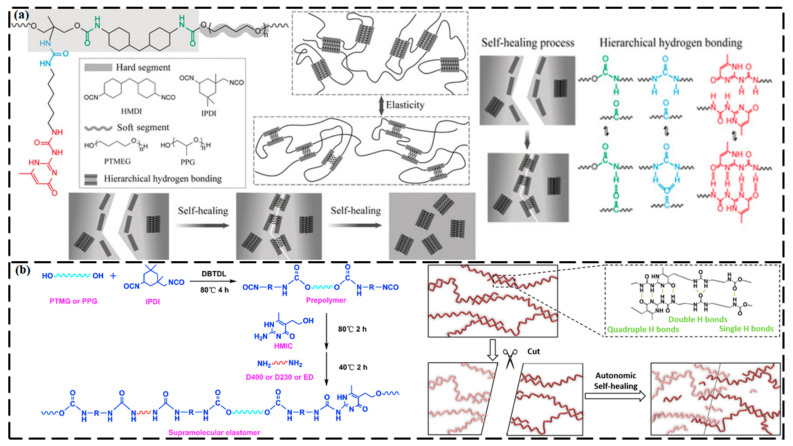
(**a**) Super-tough healable polyurethane elastomer that was designed by biomimetic hierarchical H-bonding interactions. (Reproduced with permission [88] from Wiley-VCH). (**b**) Self-healing mechanism of a supramolecular elastomer with flexible spacer units and multiple H-bonds. (Reproduced with permission [91] from Elsevier).

##### π–π Stacking

The π–π stacking interactions between a π-electron-deficient unit and a π-electron-rich unit were used in the development of thermally-activated reversible self-repairing materials [24,96]. The materials consisted of a chain-folding copolyimide with multiple π-electron-deficient receptor sites and pyrenyl π-electron-rich end-capped polysiloxane [97] or polyamide [98] as a spacer. The *T_g_* of the material could be altered by varying the spacer and the blend composition to achieve self-repair at a broad range of temperatures (approximately 50–100 °C). Due to the presence of a flexible “soft” spacer, the π–π interactions were disrupted by heating, allowing the pyrenyl end-capped chains to release from the copolyimide and flow. Thus, healing of the fracture and restoration of mechanical strength were achieved through the re-establishment of the π–π interactions. 

Burattini et al. [99] designed a supramolecular polymer composite that was based on aromatic π–π stacking and H-bonding. The polymeric composite consisted of a chain-folding polyimide, and a telechelic polyurethane with pyrenyl end groups was compatibilized by aromatic π–π stacking between the π-electron-rich pyrenyl units and the π-electron-deficient diimide groups (Figure 6). The test results showed outstanding SH efficiency after fracture, with 91% recovery of the elongation at break, 95% restoration of the tensile modulus, and 77% recovery of the toughness. Nitrobenzoxadiazole-comprising cholesterol derivatives enabled a distinct combination of intermolecular H-bonding and π–π interaction [100], where a change in the length of the spacer joining the nitrobenzoxadiazole and cholesterol units caused a dramatic change in the gel properties and healing ability. Networks with improved mechanical properties were formed by tweezer-type bispyrenyl end groups rather than monopyrenyl groups [101]. These polymers could heal at a higher temperature (~140 °C).

In an another strategy, researchers merged π–π interactions with metal–ligand interactions by integrating a cyclometalated platinum(II) complex, Pt(6-phenyl-2,2′-bipyridyl)Cl, into a polydimethylsiloxane [102]. The authors reported that the polymeric films healed at room temperature, whereas a complete regaining of the modulus of toughness and tensile modulus was achieved within 12 h. Moreover, a rise in the temperature reduced the mending time, enabling complete restoration of the mechanical properties within 2 h at 50 °C or within 1 h at 80 °C. The metal–ligand interactions were also reported to be insensitive to surface aging.

The aforementioned studies show impressive examples of π–π interactions that enable self-repairing behavior in polymeric materials. Unexplored opportunities remain for further enhancements, especially in liquid-crystalline polymers, polymers with high *T_g_*s, polymers paired with ionic liquids, and polymers with host–guest (HG) interactions or metal coordination bonds. The anisotropic nature of π–π interactions enable the development of composites with directional and reversible bonding features that are not usually observed in isotropic systems [103].

The aforementioned research shows admirable examples how π–π interactions can enable SH characteristics in polymers. There are untouched prospects for more advances, specifically, in polymers with high *T_g_*s are liquid crystalline polymers, or when combined with ionic liquid, host-guest interactions, or metal coordination. The anisotropic nature of π–π interactions may offer unique directional and reversible SH properties in materials and are usually unattainable in isotropic systems [103].

##### Metal–Ligand Coordination Bonds

Among the various possible noncovalent interactions, metal–ligand interactions are of particular interest. These interactions occur between a metal ion (called a coordination center) and the functional groups of neighboring organic molecules (called ligands) [104]. Unlike H-bonds, which have less bond energy and are sensitive to thermal aging, metal–ligand coordination bonds are more selective, reversible, and stable even at higher temperatures [105]. These coordination bonds can be used to construct self-repairing polymers because of their ability to form metallic bonds spontaneously and their bonding energies are extremely adjustable; in some cases, coordination bonds can be both thermodynamically stable and kinetically labile [68]. The strength of coordination bonds is intermediate between the strengths of van der Waals interactions and covalent bonds, and the bond’s strength can feasibly be amended by carefully selecting the combination of metal ion and ligand [1,68]. 

Burnworth et al. [106] fabricated the first self-repairing metallosupramolecular polymers. They synthesized a metallopolymer by functionalizing poly(ethylene-*co*-butylene) with 2,6-bis(1′-methylbenzimidazolyl) pyridine moieties at the termini, followed by the addition of lanthanum tri[bis(trifluoromethylsulfonyl)imide] or zinc di[bis(trifluoromethylsulfonyl)imide]. The resultant metallopolymer showed self-repairing ability when it was illuminated by ultraviolet (UV) radiation with a wavelength corresponding to the absorption band of the polymer. Shao et al. [107] designed a nanocomposite hydrogel that was crosslinked by H-bonds and dual metal–carboxylate coordination bonds (between iron ions (Fe^3+^) and carboxylic groups from poly(acrylic acid) and carboxylated cellulose nanofibrils) (Figure 7a). These hydrogels show a high fracture elongation of 1803%, a toughness of ~11.05 MJ m^−3^, a fracture strength of 1.37 MPa, and a rapid self-recovery ratio of 95.7% within 60 min and a self-repairing efficiency of 94.2% within 48 h at 25 °C; this SH performance is superior to that of the hydrogels without the ions. The authors attributed this tremendous performance primarily to the H-bonds and ionically crosslinked metal–carboxylate coordination complexes that act as sacrificial bonds to dissipate energy within the supramolecular network.

Sandmann et al. [108] synthesized a copolymer comprising of attached bidentate triazolepyridine ligands and a low-*T_g_* lauryl methacrylate backbone. The polymer was crosslinked with various salts of Fe^2+^ and Co^2+^. The as-prepared materials displayed good healing performance within time intervals of 5.5 to 26.5 h at moderate temperatures of 50–100 °C. Jia et al. [109] prepared a highly elastic polymer by incorporating dynamic Fe^3+^–triazole coordination bonds into a polydimethylsiloxane backbone. The synthesized polymer was stretched to over thirty-times its original length. When broken, the polymer was thermally healed by more than 90% within 20 h at 60 °C. The notable healing ability and mechanical properties of the synthesized polymer were credited to the unique coordination bond strength and coordination conformation of the Fe^3+^–triazole coordination complex. He et al. [110] constructed a hybrid hydrogel with an SH function by preparing a polymer that was composed of 1-methyl-3-(4-vinylbenzyl) imidazolium chloride, poly(sodium *p*-styrenesulfonate hydrate), poly(vinyl alcohol), and gold nanoparticles. The synthesized hybrid hydrogel demonstrated a high SH capability because of the reversible metal–ligand coordination between the hydroxyl groups of the polymer and the gold nanoparticles.

When coordination bonds are merged with H-bonding, synergistic effects can arise. For instance, Cui et al. [111] synthesized a self-repairing polymer via combined urea H-bonding and Zn(II)–imidazole interactions that were inserted through a bifunctional monomer of 2-(3-(3-imidazolylpropyl)ureido)ethyl acrylate. The dual dynamic effects of urea H-bonding and Zn^2+^–imidazole interactions can provide polymers that simultaneously exhibit good mechanical properties and improved self-repairing capabilities (Figure 7b). The as-prepared polymers with urea H-bonding and Zn^2+^–imidazole interaction motifs demonstrated better than 90% healing efficiency under mild conditions. When the Zn^2+^/imidazole molar ratio was varied, the mechanical strength of the synthesized material could be easily tuned from 35.0 kPa to 4.41 MPa. The self-repairing ability of the elastomer originated from the dynamic nature of the urea H-bonding, and the Zn^2+^–imidazole coordination contributed to the refurbishment of the polymeric structure. The H-bond and metal-coordinated bond should be well optimized to achieve synergistic effects in both SH performance and mechanical properties [105,112].

In another study, an elastomer was synthesized via a one-pot polycondensation reaction between bis(3-aminopropyl)-terminated poly(dimethylsiloxane) and 2,4′-tolylene diisocyanate and was subsequently coordinated with Al^3+^ ions to create metal-coordination bonds [113]. The quadruple H-bonds could not only rapidly repair damage but could also dissipate strain energy as a weak dynamic bond, providing the elastomer with outstanding stretchability and SH capability. By contrast, the triple Al-coordination bond behaved as a strong dynamic bond that considerably increased the elasticity and robustness of the elastomer. The elastomer showed an SH efficiency of 90%, along with 1700% stretchability and an impressive tensile strength of 2.6 MPa.

Metal-ligand coordination bonds are a very unique type of non-covalent bond. In most of the cases, the formation of metal bonds is spontaneous, the bond energies are highly tunable, and sometimes the bond can be both thermodynamically stable and yet kinetically labile. Such characteristics are advantageous in the construction of SH polymers. Furthermore, the presence of functional metal ions and/or ligands and the dynamic metal ligand bond can lead to a variety of functionalities, making it easy to generate functional SH polymers [68].

**Figure 7 ijms-23-00622-f007:**
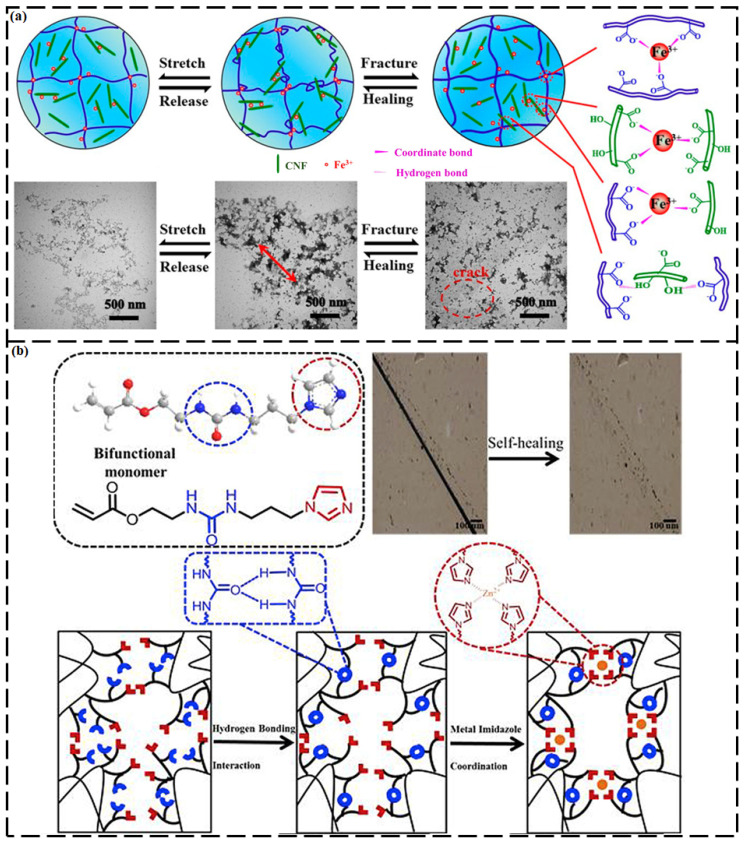
(**a**) Self-healing mechanisms of poly(acrylic acid)–cellulose nanofibrils–Fe^3+^ (PAA–CNF–Fe^3+^) by the synergy of H-bonds and dual coordinate bonds. (Reproduced with permission [107] from American Chemical Society). (**b**) Self-healing polymers that were designed via combined H-bonding and Zn–imidazole interactions. (Reproduced with permission [111] from Elsevier).

##### Host–Guest Interaction

HG interaction involves the targeted binding of two dimensionally similar but structurally different macromolecular moieties via weak interactions—specifically, π–π interactions, H-bonding, hydrophobic interactions, or van der Waals interaction. However, naturally existing HG assemblies display biological importance and synthetic HG complexes that undergo reversible guest exchange [4,114] potentially enable the synthesis of self-healable supramolecular gels. HG interactions are intriguing processes for developing biocompatible SH hydrogels. SH capability relies on the dynamics of the complexation, which entails the fabrication of energetically and geometrically suitable supramolecular networks [103].

Cyclodextrin (CD)-amended host polymers are important SHMs because of their biocompatibility, multi-stimuli responsiveness, solubility in water, low cost, and promising association/dissociation dynamics. SH by guest polymers comprising of *n*-butyl acrylate [115], ferrocene [116], *N*-vinylimidazole [117], adamantine [118,119], and *N*-adamantane-1-yl-acrylamide [120] has been reported. HG interaction between β-CD-functionalized acrylamide and ferrocene-functionalized poly(acrylic acid) chains enabled SH capability after 24 h at 20 °C [121]. Redox stimulus induced a sol–gel phase transition in the supramolecular hydrogel and facilitated the SH process between damaged surfaces. Supramolecular hydrogels that were designed from poly(acrylamide) that was integrated with CD crosslinked with aliphatic guest groups by hydrophobic interaction also demonstrated healing upon re-joining of the damaged surface [120]. Jia and Zhu [122] used β-CD and cholic acid HG interaction to prepare a self-healable supramolecular hydrogel (Figure 8). Self-healing at 20 °C was accomplished within less than one minute, and the SH time was found to depend on the cross-sectional size of the induced damage (i.e., larger areas of damage required longer healing times to achieve complete self-healing). Wei et al. [123] fabricated an HG macromer-based supramolecular hydrogel. The HG macromer hydrogel was synthesized via HG complexation between adamantine-functionalized hyaluronic acid as a guest polymer and monoacrylated β-CD as a host monomer and was used to prepare self-healable biopolymer-based freestanding supramolecular hydrogels. In one report [124], the authors constructed a supramolecular material by combining poly(methyl methacrylate) with pendent dibenzo-24-crown-8 groups and bisammonium as crosslinkers. This material was capable of 100% self-repair in less than 10 s. There are two processes that might be involved in the rapid self-repairing process: HG interaction or H-bonding and an electrostatic interaction between the crosslinker and dibenzo-24-crown-8 molecules. In another report, crown ethers were used as part of an HG pair [125]. The host polymer was constructed from glycidyl triazole polymers that were functionalized with dibenzo-24-crown-8 as a host molecule and secondary ammonium groups as a guest molecule. These materials exhibited good toughness and elasticity along with good self-repairing characteristics because of their large molecular weight and effective HG interactions. 

#### 2.2.2. Reversible Covalent Bonding

##### Cycloaddition Reactions

Cycloaddition is a particular kind of chain exchange reaction in which unsaturated species join together and form a ring [72]. The most common cycloaddition reaction is a thermo-reversible Diels–Alder reaction in which reversible crosslinking arises via [4+2] cycloaddition of a conjugated diene, which is electron-rich, and a substituted alkene, generally called the dienophile, which is electron-deficient. The bracket notation (e.g., [4+2]) symbolizes the number of electrons that are contributed by each molecule [72]. The Diels–Alder reaction is ideal for the construction of mendable polymers because it does not entail any additional chemicals, such as a catalyst. Moreover, a reversible Diels–Alder reaction does not contain free radicals, thereby limiting the side reactions that can prevent the re-formation [126].

Various SHMs such as polyamides, polyacrylates, and epoxies can be synthesized using a Diels–Alder reaction [127]. In these materials, cracking or an increase in the temperature breaks the bonds between the diene and dienophile and lowering the temperature leads to restructuring of the covalent bonds to heal the fractures [72,127,128,129]. In a recent study, a series of dynamic covalent networks that were based on the Diels–Alder reaction were synthesized by reacting linear copolymers of aliphatic methacrylates and furfuryl methacrylate with aliphatic bismaleimides. The authors obtained a colorless and extremely transparent SH polyacrylate coating [129]. Diels–Alder reactions have the disadvantage of requiring greater temperatures to trigger the SH process and high temperature exposure may not be desirable for polymers because of the *T_g_* of polymers can be within 100 °C [1,130].

To achieve self-healing of a material without extreme heating, researchers carried out photochemical cycloadditions in which an optically-induced self-repairing process was conducted at a lower temperature. These reactions include the [2+2] cycloaddition of 1,1,1-tris-(cinnamoyloxymethyl)ethane [131] or coumarin [132] and the [4+4] cycloaddition of anthracene derivatives [133]. The self-repairing phenomenon of these materials is analogous to that of the Diels–Alder reactions in which the ring opening and closure of the bonds within a polymeric structure are regulated by exposure to different wavelengths of light [131]. For example, the [2+2] cycloaddition of 1,1,1-tris-(cinnamoyloxymethyl)ethane monomers forms cyclobutane [131]. When the cyclobutane C–C bond breaks, only separate cinnamoyl groups are present. Upon exposure of the material to UV light (>280 nm), [2+2] cycloaddition repairs the bond and re-forms the cyclobutane ring (Figure 9a). An analogous [2+2] cycloaddition was reported for coumarin. Coumarin [2+2] was photodimerized at 350 nm, and the reaction was reversed at 254 nm [132]. Froimowicz et al. [133] polymerized a derivative of anthracene through a [4+4] cycloaddition reaction under UV irradiation and then explored using this process to construct self-repairing polymers. Cycloaddition reactions have been widely studied by scientists because of their good adaptability to numerous polymers and their ability to attain excellent SH capability. 

Though, several numbers of publications have been reported on multiple times self-healing capability of Diels-Alder based self-healing polymers, but the major disadvantages are poor mechanical and thermal performance [134]. Some of the reactions are summarized in Table 1.

##### Chain Exchange Reactions

Another type of reaction that is used for the development of SHMs involves covalent bonds that are capable of participating in chain exchange reactions. These reactions include the re-formation of bonds, usually between chains, sometimes within a single chain. In comparison with the supramolecular network approach, these methods benefit from the dynamic reversibility of covalent bonds. One of the great benefits of a chain exchange reaction is that it entails a lower temperature for activation of the SH mechanism than the Diels–Alder reaction [135]. Disulfide bonds tend to undergo exchange reactions in which two adjacent S–S bonds are broken and restructured via ionic intermediates or free radicals, where fracture recovery can be initiated at low temperatures via oxidation, heating, or photolysis [95,136,137,138,139,140].

Canadell et al. [141] designed a coating with commercial epoxy resins that consisted of disulfide bonds that can heal autonomously at moderate temperatures and restore the coating’s mechanical properties completely. Amamoto et al. [60] used a disulfide interchange reaction to prepare a covalently cross-linked polyurethane with excellent SH performance under visible light and ambient conditions. The amount of disulfide groups was found to determine the recuperation strength, and the mending efficiency was found to improve with increasing concentration of disulfide groups. Lai et al. [142] constructed what they refer to as a hard phase-locked self-healable polyurethane with excellent tensile stress and a high Young’s modulus of 56 MPa. They attributed the improved mechanical properties to a microphase-separated structure. The microscale aggregation of the hard segments in which the dynamic disulfide bonds were located could function as a physical crosslinker. The locked disulfide linkages were easily triggered to induce a dynamic exchange reaction at temperatures that were greater than the *T_g_*. The phase-locked polyurethane, therefore, also displayed effective self-repairing and reprocessing ability.

**Figure 9 ijms-23-00622-f009:**
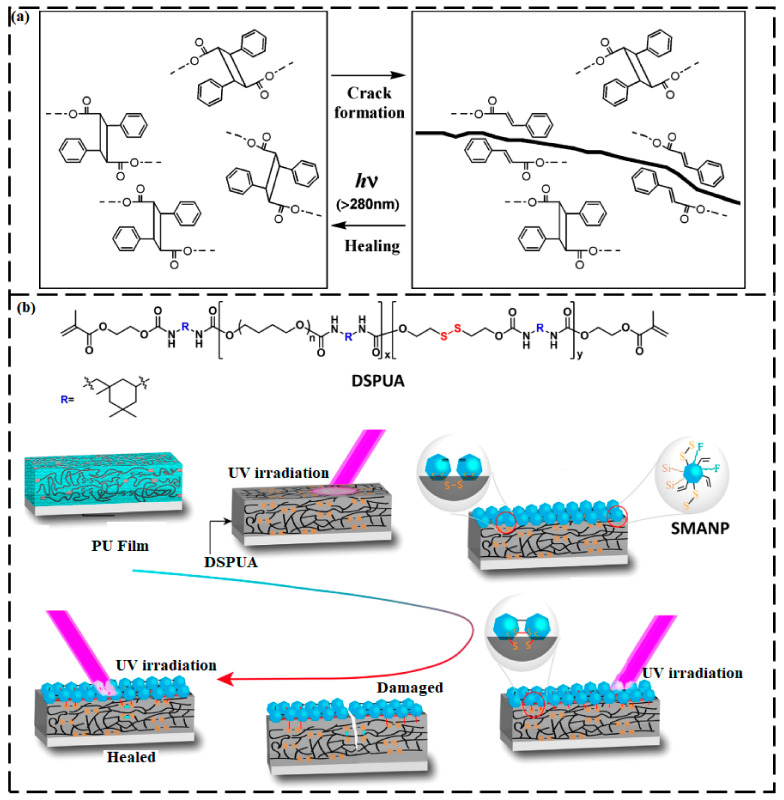
(**a**) Photochemical [2+2] cycloaddition of cinnamoyl groups. (Reproduced with permission [131] from American Chemical Society). (**b**) Photo-promoted disulfide exchange reaction for self-healing of superamphiphobic coatings. (Reproduced with permission [143] from American Chemical Society).

Zhao et al. [143] induced the formation of a disulfide bond between UV-light-curable polyurethane acrylic resins (DSPUA) and showed that deep damage in the cured resin material could be self-healed by UV irradiation (Figure 9b). Moreover, the surface of Al_2_O_3_ nanoparticles (SMANP) was amended and functionalized by hydrophobic fluorinated alkyl, vinyl, organic silane, and disulfide bonding compounds. Eventually, a superamphiphobic coating (SMANP@DSPUA) was prepared by spin-coating SMANP solution onto a DSPUA substrate, followed by curing under UV light. SMANP and DSPUA bonded covalently and formed a rough surface consisting of sturdy nano/microstructures via a disulfide exchange reaction and cross-linking of the double bond by UV curing and heating. The disulfide exchange reaction led to the restructuring of the granular layer and the nascent layer of DSPUA by covalent bonding via self-repair. The broken surface of the cracked coating could be restructured into an integrated surface, enabling the material to mend itself, even if the crack was deep. The authors also reported that increasing the disulfide bond content and the temperature could enhance the mending ability of the coatings and that UV irradiation caused a substantial increase in the healing efficiency. 

Lafont et al. [144] reported the effect of cross-linkers and chain rigidity on the self-repairing ability of a polysulfide-based thermoset material and suggested that introducing additional cross-linking points to the system would support the self-repairing of disulfide bonds. Zhang et al. [145] investigated the self-repairing mechanism of microcracks on a waterborne polyurethane with various disulfide bond contents. The exchange reaction activation energy of the disulfide bond was measured using gel permeation chromatography and found to be 20.42 kJ mol^−1^. The SH ability was found to be influenced by the amount of disulfide bonds, temperature, external force, and the microstructures. The healing rate of the synthesized waterborne polyurethane was reported to be 100% at 75 °C within 15 min. 

**Table 1 ijms-23-00622-t001:** Reported SHMs with their healing mechanism, healing efficiency, and healing conditions.

Mechanism	Materials	Healing Conditions	Healing Efficiency and Tensile Strength	References
Ionic interaction	Poly(acrylic acid)/poly(allylamine hydrochloride) compact polyelectrolyte complexes	Room temperature, 8 h, 1 M NaCl	approx. 100%	[74]
	Imidazolium-containing photocurable monomer, 6-(3-(3(2-hydroxyethyl)-1H-imidazol-3-ium bromide)propanoyloxy)hexyl acrylate, isobornyl acrylate, 2-(2- ethoxyethoxy)ethyl acrylate, and 2-hydroxyethyl acrylate	Room temperature to 120 °C, 24 h	93% and 3.1 MPa	[76]
	Alginate-based supramolecular cationic polyurethanes	-	87.3% and 48 MPa	[77]
	Zwitterionic polyurethanes from *N*,*N*-bis(2-hydroxylethyl) isonicotinamide, hexamethylene diisocyanate, and 1,3-propanesultone	60% RH, room temperature	100%	[79]
	Zwitterionic multi-shape-memory polyurethanes (ZSMPUs) from *N*-methyldiethanolamine, hexamethylene diisocyanate and 1,3-propanesultone	-	-	[78]
Hydrogen bonding	PSeD-U bioelastomers synthesized by grafting 2-ureido-4[1H]-pyrimidinones (UPy) and poly(sebacoyl diglyceride) (PSeD)	60 °C, 30 min	100% and 1.88 MPa	[89]
	Self-healing polyurethane (SPUs), prepared by grafting 2-ureido-4-pyrimidone moieties in castor oil-derived polyurethane	60 °C, 24 h	84% and 1.8 MPa	[90]
	Supramolecular elastomers synthesized from polypropylene glycol and polyetheramines (230)	Room temperature, 24 h	4.18 MPa	[91]
	Supramolecular elastomers synthesized from polypropylene glycol and polyetheramines (400)	Room temperature, 24 h	90% and 6.27 MPa	[91]
	Supramolecular elastomers synthesized from poly(oxytetramethylene) glycol and polyetheramines (400)	Room temperature, 24 h	4.77 MPa	[91]
	Supramolecular elastomers synthesized from polypropylene glycol and ethanediamine	Room temperature, 24 h	3.61 MPa	[91]
	Supramolecular hydrogel of chitosan in the presence of graphene oxide nanosheets as cross-linkers	1 min	Same mechanical property as original one	[92]
	Electrodes of supercapacitor are fabricated by spreading functionalized single-walled carbon nanotube	50 °C, 5 min	100%	[93]
π–π stacking	Cyclometalated platinum(II) complex/polydimethylsiloxane	12 h, room temperature	Almost 100%	[102]
	Azobenzene-containing liquid crystalline polyester	60 °C, 5 h	73.5% and 12.8 MPa	[146]
π–π stacking and H-bonding	Supramolecular polymer blend based on polyimide and a telechelic polyurethane with pyrenyl end groups	100 °C, 240 min	3 × 10^5^ Pa	[99]
Metal–ligand coordination bonds	Metallopolymer synthesized by poly(ethylene-*co*-butylene) with 2,6-bis(1′-methylbenzimidazolyl) pyridine moieties at the termini, followed by the addition of zinc di[bis(trifluoromethylsulfonyl)imide]	-	100%	[106]
	Metallopolymer synthesized by poly(ethylene-*co*-butylene) with 2,6-bis(1′-methylbenzimidazolyl) pyridine moieties at the termini, followed by the addition of lanthanum tri[bis(trifluoromethylsulfonyl)imide]	-	104%	[106]
	Fe(III) coordinated with triazole/polydimethylsiloxane	60 °C, 20 h	Over 90%	[109]
	Bis(3-aminopropyl)-terminated poly- (dimethylsiloxane)/ 2,4′-tolylene diisocyanate/Al(III) ions	36 h, room temperature	90% and 2.6 MPA	[113]
	Ni^2+^ coordination of polyethylene glycol with bistriazole pyridine ligands	2 min, room temperature	100%	[147]
Hydrogen-bonding and metal–ligand coordination bonds	Poly(acrylic acid)- cellulose nanofibrils- iron ions (Fe^3+^) gels	25 °C, 48 h	94.2%	[107]
	1-methyl-3-(4-vinylbenzyl) imidazolium chloride/poly(sodium *p*-styrenesulfonate hydrate)/poly(vinyl alcohol)/gold nanoparticles	-	-	[110]
Host–guest interaction	β-cyclodextrin and cholic acid	20 °C, 1 min	-	[122]
	Adamantine-functionalized hyaluronic acid/β-cyclodextrin	2 min	-	[123]
	Poly(methyl methacrylate) combined with pendent dibenzo-24-crown-8 groups and bisammonium as crosslinkers	10 s	100%	[124]
Diels–Alder reactions	Polyacrylate coating produced by reaction of furfuryl methacrylate and bismaleimides	150 °C (90 min) then cooling at room temperature	>90%	[129]
Photochemical [2+2] cycloaddition	Dihydroxyl coumarin based polyurethane	254 nm UV then 350 nm UV, 42 h	64.4%	[132]
	Cinnamoyl mechanophore	48 h UV irradiation	24%	[148]
Disulfide (S–S) exchange reactions	Superamphiphobic coatings, fabricated by deposition of Al_2_O_3_ nanoparticles into polyurethane acrylic resin	80 °C, 90 min	Complete (almost 100%) removal of scratch and 12.8 to 16.5	[143]
	Waterborne polyurethane	Room temperature, 24 h	95.18%	[149]
	Waterborne polyurethan	75 °C, 15 min	100%, 100 MPa	[145]

## 3. Applications of Self-Healing Materials

### 3.1. Energy-Harvesting Devices

#### 3.1.1. Perovskite Solar Cells

Perovskite solar cells (PSCs) have triggered a wave of development of thin-film photovoltaics with a power conversion efficiency from 3.8% to 25.2% [150,151]; this high conversion efficiency is attributed to the small exciton binding energy, high absorption coefficient, and long carrier diffusion length of their perovskite-structured phase [152,153]. Although a satisfactory power conversion efficiency has been achieved, poor long-term stability remains a core problem and impedes the further development of PSCs. The degradation of PSCs is attributable to the volatility and hygroscopicity of the organic constituents that are aggravated by external stimuli (e.g., oxygen, UV light, and moisture) [154,155]. Thus far, researchers have devoted extra attention to prolonging the stability of PSCs under actual operating conditions. The latest improvements in SH technology have provided a favorable approach for improving the long-term stability of PSCs [151,156].

Zhao et al. [157] designed a self-repairing lead halide PSC in which polyethylene glycol (PEG) was chosen as the polymer scaffold material and was added to a precursor solution of PbCl_2_ and CH_3_NH_3_I (MAI) to form a continuous PEG scaffold perovskite absorber layer via a mild-temperature process (Figure 10a). The improved stability and self-repairing ability of PSC devices were attributed to the hygroscopicity of the PEG molecules and their robust binding with MAI, which prevented MAI from evaporating. These PEG molecules could efficiently absorb water to form a compact moisture barrier around the perovskite crystal grains, thereby causing the penetration of a small amount of water into the perovskite film. When sprayed with water, the MAPbI_3_ perovskite-structured crystals decomposed into MAI and PbI_2_. The MAI molecules were anchored by the adjacent PEG molecules through robust bonding between the MAI and PEG. In the absence of water vapor, PbI_2_ within the film reacted again with the anchored MAI molecules and re-formed the perovskite MAPbI_3_. This self-repairing ability was attributed to the rapid decomposition–regeneration process of the perovskite-structured crystals. Upon contacting water vapor for 60 s, the perovskite-type film with PEG first turned from black to yellow and then recovered to black within 45 s after the water vapor was removed. By contrast, the perovskite-type film without PEG turned yellow irreversibly. The PSCs exhibited efficiencies as high as 16%, as well as a high tolerance to humidity and excellent SH characteristics. Thermal degradation of PSCs is another problem that limits their long-term stability. Fan et al. [158] incorporated 2-(1*H*-pyrazol-1-yl)pyridine into 3D Cs_0.04_MA_0.16_ FA_0.8_PbI_0.85_Br_0.15_ (Cs: cesium, MA: methylammonium, FA: formamidinium, Pb: lead, I: iodine, Br: bromine) perovskites to construct a series of 1D and/or 1D–3D lead halide hybrid designs. The fabricated 1D–3D perovskites demonstrated excellent thermodynamic self-repairing performance and good long-term thermal stability. In a recent work [151], a mechanically strong and self-repairable thiourea–triethylene glycol polymer with a *T_g_* near room temperature and strong interactions with perovskites was designed and incorporated into perovskite-type thin films. The thiourea–triethylene glycol polymer in a hybrid halide perovskite-type film enabled the re-joining of distant perovskite-structured grains and damage repair. A film with a polymer concentration of ≥2.1% was reported to heal a 6 µm wide scratch when annealed for 1 h at 100 °C under a N_2_ atmosphere. These results foreshadow the application of polymer–perovskite hybrid materials in ultra-flexible and wearable energy-harvesting devices.

Another major shortcoming of perovskite-based solar cells is their lack of photostability under solar irradiation [159,160,161,162]. Nie et al. [163] confirmed that the degradation in the performance of a hybrid PSC (Figure 10b) under continuous solar illumination was due to the degradation of the photocurrent; their hybrid PSC could achieve fast and complete self-repair when rested in the dark for less than one minute. The power conversion efficiency of the hybrid PSC device decreased after two hours of constant solar illumination; however, its original performance was restored after the device was rested in the dark. The authors attributed the decrease in the photocurrent to the increase in the amount of light-activated meta-stable states that could form during long-term light soaking, which resulted in the generation of charged areas in the bulk of the thin film. The light-activated trap states disappeared when the device was left in the dark and on re-illumination the photocurrent self-heals to nearly 100% of its initial steady-state value. The photocurrent deterioration and SH could be repeated for numerous cycles of operation, enabling a more effective method of managing the overall degradation rates by modifying the solar-cell operating parameters. 

Device encapsulation by polymer-based materials forming protective coatings has proven to be an extremely successful approach for enhancing the stability of PSCs. Numerous customized encapsulation techniques have been established to inhibit the diffusion of moisture and oxygen into the perovskite-type layer. This approach has been extensively used to improve the stability of solar cells against external effects. Encapsulating coatings that are composed of rigid or flexible organic polymeric materials such as polyethylene terephthalate [164], polyvinyl alcohol [165], or a multifunctional fluorinated photopolymer [166] have been used to protect PSC devices from damage under ambient conditions. However, these coatings may themselves be prone to the formation of cracks or microstretches upon prolonged exposure to harsh environmental conditions, eventually allowing moisture to pass through the coating and ultimately damage the device [167,168,169]. Banerjee et al. [167] constructed a polyisobutylene-based SH polymer as a smart coating material for photovoltaic devices. Their coumarin-functionalized polyisobutylene tri-arm star polymers underwent fast and efficient reversible cross-linking upon irradiation with UV light and the photocleavage/photodimerization cycle could be repeated numerous times without the loss of the repairability. Damage could be repaired even under exposure to sunlight, albeit more slowly. The cross-linked polyisobutylene films exhibited good oxygen and moisture barrier properties, making them potentially suitable as a coating for photovoltaic systems. The polyisobutylene-encapsulated photovoltaic devices (Figure 10c) retained 85% of their efficiency even after 20 days, whereas the efficiency of the devices with no encapsulation deteriorated to 10% of their maximum value within two days [170]. Notably, compared with the high-molecular-weight polymers, the low-molecular-weight polymers displayed enhanced stability due to their higher cross-linking density during photodimerization.

**Figure 10 ijms-23-00622-f010:**
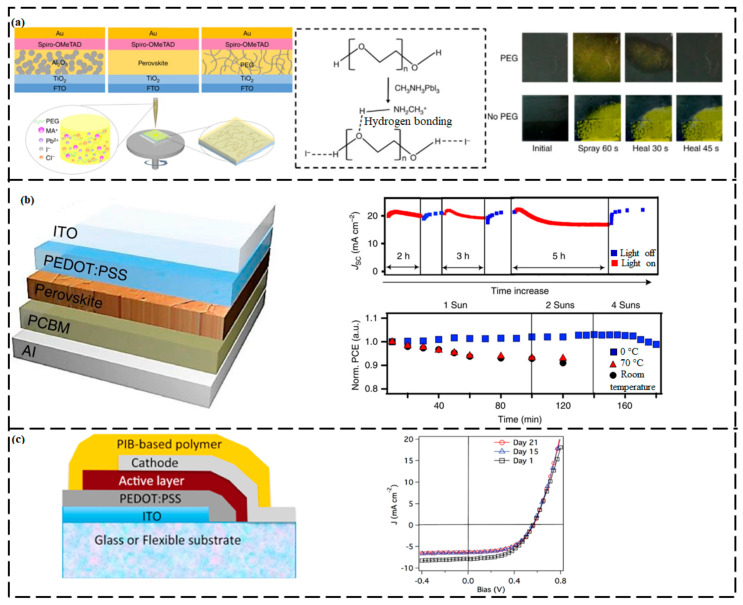
Self-healing PSCs. (**a**) PEG-facilitated SHP scaffold PSC. (Reproduced with permission [157] from Springer Nature). (**b**) Planar solar cell design. (Reproduced with permission [163] from Springer Nature). (**c**) Encapsulated PIB-based polymer solar cell. (Reproduced with permission [170] from Elsevier).

#### 3.1.2. Triboelectric Nanogenerators

Triboelectric nanogenerators (TENGs) have recently attracted intensive attention as a green and sustainable power source because of their benefits, which include low cost, ease of manufacture, vast material availability, high conversion efficiency, and diverse designability [171,172,173,174,175,176]. TENGs, which are based on the effects of triboelectrification and electrostatic induction, have been shown to be an effective alternative for converting ubiquitous ambient mechanical energies into electricity, especially for mechanical motions at low frequencies [177,178]. In conventional TENGs, patterned triboelectric layers with micro-nanostructures were used to increase the output power by enhancing the contact surface area [179]. However, overall performance loss and the degradation and short lifetime of the TENG devices remain problems and might arise from the interface friction between different materials and frequent mechanical deformation such as device twisting, stretching, bending, and compressing [180,181]. The most viable option would be to impart a SH capability to the TENGs, thereby enabling the broken micro-nanostructure of the triboelectric layer to be repaired and the overall nanogenerators performance to be recovered [18]. 

A shape-memory polyurethane-based SH TENG can be repaired and its performance can be restored after degeneration of its triboelectric layer [180]. After the TENGs were heated above the *T_g_* of polyurethane for 30 s, they recovered their original micropatterned structure and mechanical harvesting capacity. In addition, the authors checked the stability and repeatability of this healing mechanism and carried out 30 successive degradation–healing cycles on a fabricated TENG; they found that the healing worked perfectly after strong, rapid, and intense degradation cycles.

A highly stretchable and healable TENG (Figure 11a) was developed using polyurethane acrylate elastomer as the triboelectric layer and also as the polymer matrix for the conductor (comprising a liquid metal, silver flakes, and polyurethane acrylate) [182]. Polyurethane acrylate has supramolecular H-bonding, resulting in superb stretchability of 2500%, and imparts the TENGs with recovery ability after mechanical damage. After being mended, the polyurethane acrylate recovered 45.1% of its mechanical performance and 96% of its electrical conductivity. 

Sun et al. [178] designed a fast self-healable, stretchable, and transparent TENG that was based on an Ag nanowire/poly(3,4-ethlenedioxythiophene) composite as an electrode layer and poly(dimethylsiloxane) as an SH film (Figure 11b). Dynamic covalent bonds have been introduced into poly(dimethylsiloxane) networks as self-repairing agents. After the injury, imine bonds were hydrolyzed, resulting in the formation of an amine and an aldehyde. During the repairing process, imine bonds again formed and cured the injury. The TENGs displayed complete self-healing and effective energy generation even after mechanical damage and mending cycles. In this work, the resultant TENGs exhibited only 50% stretchability and 73% transmittance and required 12 h for healing and restoration of their power generation ability (100% healing efficiency) even after accidental cutting.

A fully self-healable TENG was obtained by coating a layer of polydimethylsiloxane–polyurethane onto an electrode comprising of small magnets (Figure 11c) [183]. When the TENG was damaged, the mechanical properties of the healable polymer and the electrical properties of the magnetic-assisted electrodes were self-recovered when the broken ends were simply reconnected. Both the output voltage and output current of the self-repaired TENGs recovered to more than 95% of their initial values even after the fifth cutting–healing cycle. However, the healed TENGs were neither transparent nor stretchable and required heating for recovery.

Lai et al. [184] designed an energy-harvesting TENG in which both the triboelectrically charged layer and the electrode of the TENG were completely and autonomously self-healed under environmental conditions. The designed TENG was completely self-healed within 30 min under extreme strain (900%), its transparency reached to 88.6%, and it exhibited inherent super-stretchability. It was composed of a metal-coordinated polymer as a triboelectrically charged layer and an H-bonded ionic gel as an electrode. Even after 500 cutting and healing cycles, or at an ultimate stress of 900%, the TENG retained its functionality.

A mechanically robust and SH poly(hindered urea) (PHU) network was designed, which is not only flexible but also has a high tensile strength (1.7 MPa at break) [185]. The as-synthesized material can be SH rapidly, repetitively, and re-processable under mild conditions. The authors used PHU to design SH TENGs that can regain their triboelectric performance when the damaged surfaces have fully healed. The SH TENG displayed the best triboelectric output performance (169.9 V/cm^2^) among all the reported healable TENGs for the interfacial polarization-induced enhancement of the dielectric constant

### 3.2. Energy Storage Devices

#### 3.2.1. Supercapacitors

SCs are energy-storage devices that share mechanistic aspects with both conventional capacitors and rechargeable batteries. SCs offer numerous benefits, such as excellent energy density, rapid charge and discharge characteristics, good safety, long life cycles, easy maintenance, and good stability [186,187]. SCs can deliver higher power and store more energy than conventional capacitors [188]. However, their service life can be shortened by mechanical damage. Thus, mechanically and electrically, SH SCs have attracted broad research interest.

A flexible and self-healable SC with high energy density during low-temperature operation was designed using polyampholyte and biochar as the base materials for the gel electrolyte and the electrode, respectively (Figure 12a) [189]. A polyampholyte is a tough hydrogel that offers SH ability along with preferred mechanical properties. Biochar, which is obtained from the low-temperature pyrolysis of biological wastes along with the insertion of reduced graphene oxide (rGO), is a carbon material that exhibits high electrical conductivity and good mechanical strength. The fabricated SC displayed a high energy density of 30 Wh kg^−1^ with 90% capacitance retention after 5000 charge–discharge cycles at a power density of 50 W kg^−1^ at room temperature. At a lower temperature (−30 °C), the SC exhibited an energy density of 10.5 Wh kg^−1^ at a power density of 500 W kg^−1^. This excellent low-temperature behavior is likely related to the presence of non-freezable water near the hydrophilic polymer chains.

Yeu et al. [190] fabricated and assembled self-healable micro-SCs using three-dimensional MXene–rGO composite aerogels as an electrode material (Figure 12b). The MXene–rGO composite aerogel electrode was synthesized via simple freeze-drying and laser cutting methods. An SH carboxylated polyurethane shell was also wrapped around the 3D-structured electrode. Such micro-SCs demonstrated a large specific area and a capacitance of 34.6 mF cm^−2^ at a scan rate 1 mV s^−1^. The synthesized micro-SCs also demonstrated excellent SH performance, maintaining 81.7% of their specific capacitance after five healing processes. In one report [191], an SH stretchable carboxylated polyurethane material was coated onto spring-like rGO-based composite fiber electrodes to confirm the stretchability and SH properties of the SC (Figure 12c). The fabricated SC showed 82.4% capacitance retention after a large stretch (100%) and 54.2% capacitance retention after the third healing process.

Wang et al. [192] constructed an omni-healable SC (Figure 12d) that not only spontaneously healed the electrodes but also mended the electrolyte and even healed the interface between the electrolyte and the electrode. Its healing functionality was achieved by a dynamic network that was based on poly (vinyl alcohol) cross-linked by diol–borate ester linkages. The SC demonstrated fast repair of its configuration, capacitive performance, and mechanical properties during all 15 breaking–healing cycles without an external stimulus, irrespective of the breaking sites. Tension experiments revealed that the SC could restore more than 98.5% of its stress after healing for only three minutes. Excitingly, the SC could also be designed into different small geometric patterns, which could be healed into new shapes without affecting the SC’s electrochemical performance.

#### 3.2.2. Lithium-Ion Batteries

Lithium-ion batteries (LIBs) are rechargeable batteries (secondary batteries) that generally operate by the diffusion of Li^+^ ions between a positive and negative electrode. LIBs are becoming the leading energy storage devices for portable electronics (e.g., electric vehicles, notebook computers, smartphones, wearable electronics, and power tools) because of their environmental benignancy, long lifespan, gravimetric energy density, and high volumetric energy density [193,194,195,196]. From a microscopic viewpoint, electrode materials that suffer from changes in volume and stress generally split during repetitive charge/discharge cycling. Macroscopically, the batteries certainly incur mechanical damage and abrasion such as drilling, bending, cutting, and rolling during long-term utilization [197]. Moreover, the low thickness of the LIBs, which ensures high flexibility, can also lead to damage by accidental cutting. Such damage results in the failure of LIBs and can cause severe safety problems [198]. All of these factors strongly influence the lifecycle of flexible batteries [196]. Hence, the development of both macroscopically and microscopically self-healable LIBs is important for improving their lifecycle and cycle stability [45,198].

The first flexible and self-healable LIB (Figure 13a) was reported by Zhao et al. [198]. They designed the electrodes by aligning carbon nanotube sheets that were loaded with LiMn_2_O_4_ and LiTi_2_(PO_4_)_3_ nanoparticles onto an SH polymeric substrate, and a newly developed lithium sulfate/sodium carboxymethylcellulose acted as both a gel electrolyte and a separator. The normal functionality of the as-prepared LIB could be repaired by simply bringing the two cut parts into contact for a few seconds. The specific capacity was reduced from 28.2 to 17.2 mAh g^−1^ at a current density of 0.5 A g^−1^ after the fifth cutting. Moreover, the SH LIB also demonstrated excellent mechanical strength. 

Deshpande et al. [199] proposed an approach to prepare electrode materials with both a high capacity and enhanced durability using low-melting-point metallic alloys. They examined the reversibility of lithiation of the liquid-metal pure Ga at 40 °C as a negative electrode for an LIB. Upon full lithiation (lithium insertion), it hosted two Li atoms per Ga atom and demonstrated a high theoretical capacity of 769 mAh g^−1^. Upon lithiation, the liquid-metal electrode crystallized. The solid-state Ga electrode was returned back to its liquid state after delithiation, remarkably repairing the fractures that appeared on the electrode during delithiation. Wu et al. [200] fabricated an SH anode for LIBs by using a room-temperature liquid metal. This liquid metal was an alloy of Sn and Ga, and it was stabilized in a rGO/carbon nanotube skeleton. Sn was used to decrease the melting point of Ga; the Ga–Sn alloy thus demonstrated SH ability at room temperature. In addition, the rGO/carbon nanotube skeleton increased the electrical conductivity of the electrode and obstructed the liquid metal alloy from aggregating or detaching during repetitive cycling. The smooth surface of the liquid metal alloy became rougher when charging and discharging were carried out. Upon full lithiation, the Ga–Sn became a solid with a remarkable increase in surface roughness, which the authors attributed to a volume expansion. After delithiation, the rough surface became smooth again. This behavior avoided expansion/contraction-induced fracture, thereby endowing the Ga–Sn with good cycling performance and remarkable cycling stability. 

Munaoka et al. [193] added PEG groups to an SH polymer to enable Li^+^ conduction within the binder. This concept involves improving the interface between the Si microparticles and electrolytes after cycling via the combination of rapid Li^+^ conduction and SH capability. The authors reported that this binder exhibited self-repairing capabilities because of the presence of dynamic H-bonds. In addition, the binder introduced a polyether unit that facilitated the conduction of Li^+^ ions within the binder. The authors also reported that, when the SH polymer was mixed with PEG (*M*_w_ 750) in an optimal ratio of 60:40 (mol%), the resultant composite exhibited a high capacity, reasonable rate performance, and long cycle stability. 

Gendensuren and Oh [201] attempted to improve the electrochemical performance of batteries by dual crosslinking bio-derived alginate with polyacrylamide (Figure 13b). The dual-crosslinked electrode did not exhibit obvious delamination between the electrode layer and the current collector. After 100 cycles, all the electrodes exhibited visible cracks; the non-crosslinked electrodes, in particular, exhibited more severe cracking than the crosslinked electrodes. This result was consistent with the observed cycling performance and was attributed to the mechanical robustness and SH process of the dual-crosslinked electrodes after damage.

A double-wrapped binder was fabricated for an Si anode to enable the progressive dispersion of stresses. Rigid polyacrylic acid (PAA) that behaves as a protective layer was used to dispel the inner stress during lithiation and an elastic binder bifunctional polyurethane (BFPU) with SH capability was filled into the Si electrode to act as a buffer layer to dissipate residual stress, thereby avoiding structural degradation of the rigid polyacrylic acid [202]. Polyurethane with rapid SH ability was added to mend the microcracks that were induced by large stress and to further protect the integrity of the Si anode. This multifunctional binder with a smart design of a double-wrapped structure (Figure 13c) offers insights for increasing the cycling life of high-energy-density LIBs that undergo large volume changes during the cycling process.

**Figure 13 ijms-23-00622-f013:**
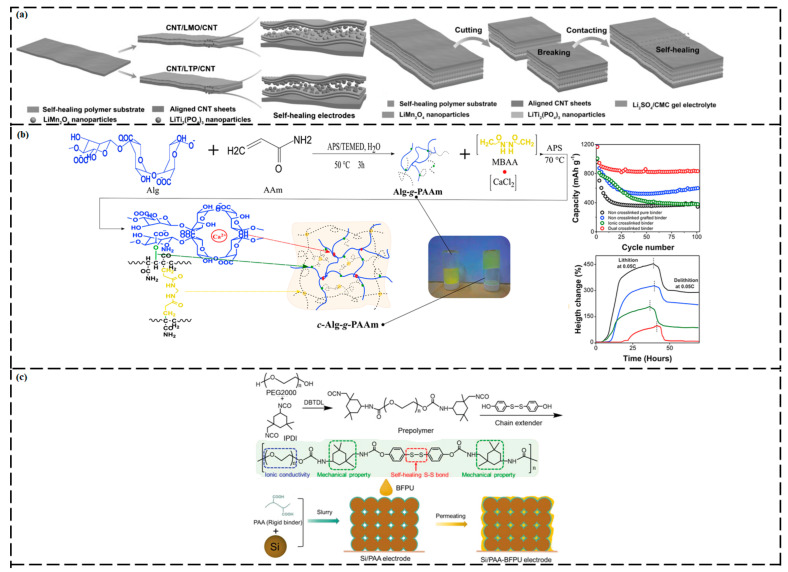
Self-healing LIBs. (**a**) All-solid-state self-healing aqueous LIB. (Reproduced with permission [198] from Wiley-VCH). (**b**) Alginate-grafted polyacrylamide for Si/graphite anodes of LIBs. (Reproduced with permission [201] from Elsevier). (**c**) Fabrication procedure of the Si anode with double-wrapped polyacrylic acid–bifunctional polyurethane binder. (Reproduced with permission [202] from Wiley-VCH).

### 3.3. Sensors

Flexible and wearable smart electronic devices have received a myriad of research interest because of their important applications in sports performance monitoring, healthcare diagnosis, human activity detection, and entertainment [203,204,205,206,207,208,209,210]. Nonetheless, sensors that respond to mechanical deformation that induces a change in the capacitance or resistance are certainly prone to microfissures or mechanical damage under repetitive deformation. The performance of the sensor is susceptible to this structural damage which results in a loss of sensor functionality. Thus, an ideal sensor should have repeatable self-healing properties to improve its lifespan and reliability while preserving its sensitivity after cutting–healing cycles [211].

Wu et al. [212] fabricated ultrasensitive and stretchable temperature sensors that were based on self-healable organohydrogels that were prepared from ethylene glycol and glycerol. The synthesized organohydrogels were stable from −18 to 70 °C and displayed an excellent freezing and drying tolerance. The organohydrogel demonstrated a remarkably high thermal sensitivity of 19.6% °C^−1^. In addition, the sensor also displayed good stretchability (1103% strain), high SH capability, and high transparency.

Wang et al. [213] fabricated a self-healable ternary polymer composite for ultrasensitive strain and pressure sensing (Figure 14a). This ternary composite comprised of polyaniline, polyacrylic acid, and phytic acid. Polyacrylic acid functioned as the soft counterpart for the rigid aromatic backbone of polyaniline chains, and phytic acid molecules acted as dopants to supply auxiliary physical crosslinking points. The as-synthesized composite displayed excellent electrical and mechanical SH properties with ~99% recovery efficiency within 24 h under ambient conditions because of the dynamic interaction between the electrostatic binding and H-bonding. The composite also demonstrated high stretchability (460%) and an electrical conductivity of 0.12 S cm^−1^. The authors also reported that, when this composite was incorporated into a pressure sensor, the sensitivity reached 37.6 kPa^−1^ in the pressure range 0–0.8 kPa and 1.9 kPa^−1^ at pressures that were greater than 5 kPa, indicating that the sensor based on this composite was suitable for sensing a wide range of tactile events. The fabricated pressure sensor was used to monitor and detect various human motions during speaking and breathing. Moreover, the composite could be rolled into a tubular shape. In this configuration, the material behaved as a tubular pressure sensor that could precisely and repeatedly measure variations in air pressure.

Cao et al. [81] used bio-derived carboxyl cellulose nanocrystals to induce multiple H-bonding interactions with chitosan-decorated epoxy natural rubber latex. The nanostructured supramolecular sensor (Figure 14b) demonstrated ultrafast (within 15 s) and repeatable SH capability, with superior healing efficiency (93% after the third healing cycle). A strain sensor was designed by the layer-by-layer technique and with H-bonding between chitosan solutions and nanocomposite-assisted carbon nanotubes. The sensor showed a low strain detection limit of 0.2% with stable and repeated response signals, even after cutting–healing and after being bent more than 20,000 times. Moreover, the authors fabricated a human–machine interaction system by integrating sensitive and self-healable and flexible sensors and subsequently used the system as a facial expression control system and an electronic larynx.

Wang et al. [214] prepared zwitterionic nanocomposite hydrogels that were physically crosslinked by exfoliated laponite-XLG nanosheets with compliant adhesion, high strain sensitivity, reliable self-healing, and good mechanical properties and used the crosslinked hydrogels for skin strain sensors used to monitor body motion (Figure 14c). The reversible physical interaction imparted the hydrogels with fast SH capability. After 24 h, the repaired hydrogel was stretched to 0.17 MPa and 1700% and demonstrated 74% fracture-strength retention.

## 4. Conclusions and Future Outlook

The ability of materials to mimic the SH attributes of nature has been a field of growing interest, and remarkable progress has been made over the past few decades in extending the lifecycle of devices and improving the utilization of energy and resources. This article reviews recent and intriguing advances in the field of SH polymeric materials and composites as well as their prospective applications in electronic devices. The application of SHMs in the design and development of electronic devices indicates that the addition of SH ability to materials is a viable step toward improving the reliability and durability of electronic devices.

A large variety of chemistries that are related to both the extrinsic and intrinsic SH methods have been explored recently. The extrinsic method includes introducing external healing agents into the system, whereas the intrinsic mechanism refers to the inherent reversibility of the molecular interaction of the polymeric matrix which is initiated by external stimuli. However, the extrinsic SH approach enables only one-time healing, whereas the intrinsic method facilitates repetitive repairing. Thus, the intrinsic SH process is far more advantageous because it not only extends the material’s lifecycle but also provides extended protection.

For both intrinsic and extrinsic SH systems, finding high throughput, scalable, and economical techniques in mass production of these materials needs intense research studies. However, recent research has been involved more on the fabrication and design of SH materials for specific purposes and cases rather than commercialization and mass production. Enhancing the properties of virgin SHM is crucial, as SHM usually have lower performance compared to their conventional versions. The application of the SH concept in practical applications under various conditions (pressure, light, temperature, humidity, vacuum, etc.) remains a challenge, and this must be carefully considered during the design phase of SHMs to achieve not only long-term but also high healing performance throughout its service life. The great majority of on-going research on SHMs will and should surely overcome these obstacles and lead to the development of tailor-made SHMs for variety of applications.

The repetitive rapid SH, compatibility, flexibility of materials, and stable metallic conductivity after damage are extremely important features that require further investigation. To attain the original performance, the SH efficiency needs to be improved. For SH conductive materials, the healing efficiency of electrical conductivity and mechanical behavior is contradictory; therefore, how to achieve the synergistic characteristics of high conductivity and mechanical properties needs to be studied in the future. In the case of some of the challenges (e.g., the incorporation of all appropriate SH components into a single device to produce entirely SH devices, and non-autonomic mending conditions of materials and devices), the performance of existing SH polymers remains inferior to that of nonhealing materials, which restricts their eventual performance and use in devices. To overcome these issues, multiscale theoretical simulations and modeling are needed to elucidate the repair mechanisms at the molecular level. With deeper insights at the nanoscale-to-microscale level, new concepts, theories, and scalable fabrication techniques can be devised for the preparation of novel materials.

Future devices that can self-heal repeatedly and quickly in typical working conditions are expected. Materials that self-repair under ambient conditions are particularly useful for the creation of smart SH electronic devices. As self-repairing materials continue to evolve at a rapid pace, devices that are fashioned from SHMs will eventually be able to function in harsher working conditions, where tedious device maintenance operations can be obviated by the use of self-repairing materials. Innovations in this research field could pave the way for devices with better efficiency and strength and with less maintenance and replacement overheads. Importantly, through their autonomic repair or semi-autonomic activities, these SHMs tend to decrease energy consumption and waste generation and propel our connected society toward a more sustainable future.

## Figures and Tables

**Figure 1 ijms-23-00622-f001:**
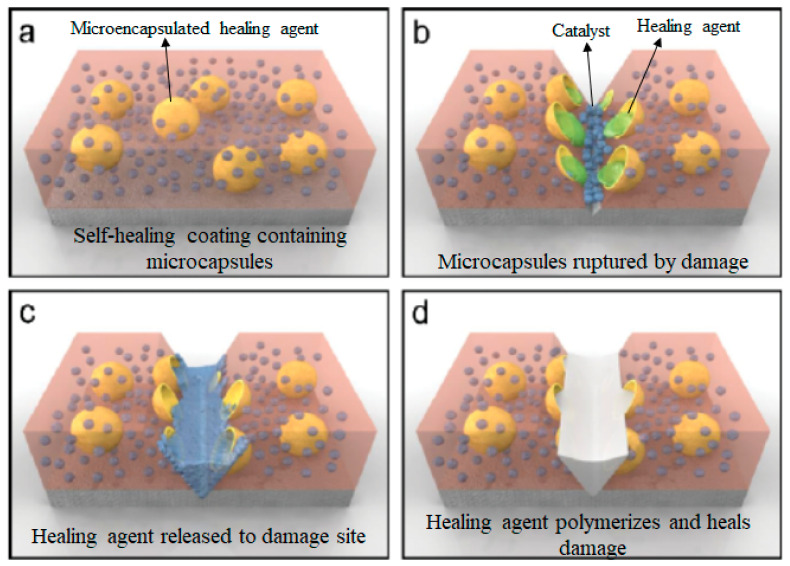
Representation of the SH concept using embedded microcapsules. (**a**) SH coating containing microencapsulated catalyst (yellow) and phase-separated healing agent droplets (blue) in a matrix (light orange) on a metallic substrate (grey) (**b**) Damage to the coating layer releases catalyst (green) and healing agent (blue) (**c**) Mixing of healing agent and catalyst in the damaged region (**d**) Damage healing. (Reproduced with permission [29] from Wiley-VCH).

**Figure 2 ijms-23-00622-f002:**
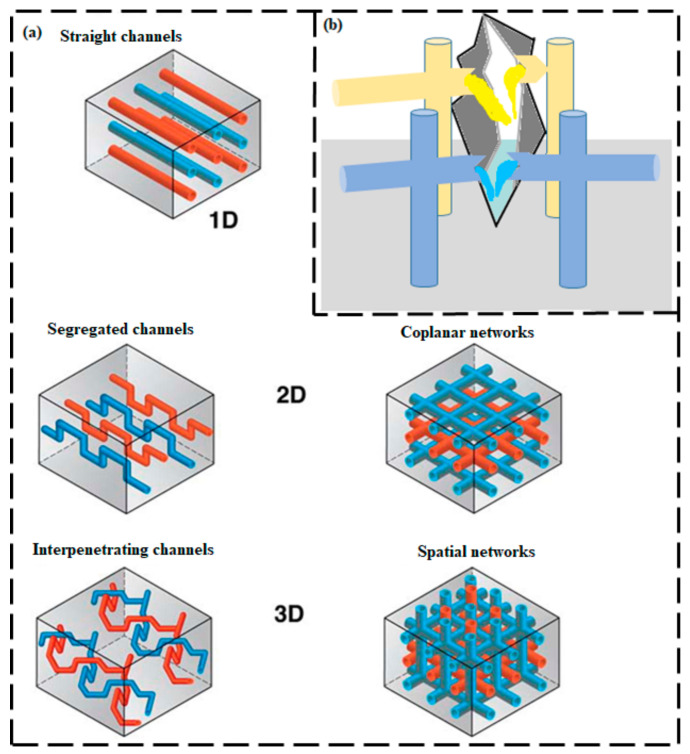
(**a**) Different types of vascular networks. (Reproduced with permission [42] from Elsevier). (**b**) Representation of the self-healing concept using microvascular embedment.

**Figure 3 ijms-23-00622-f003:**
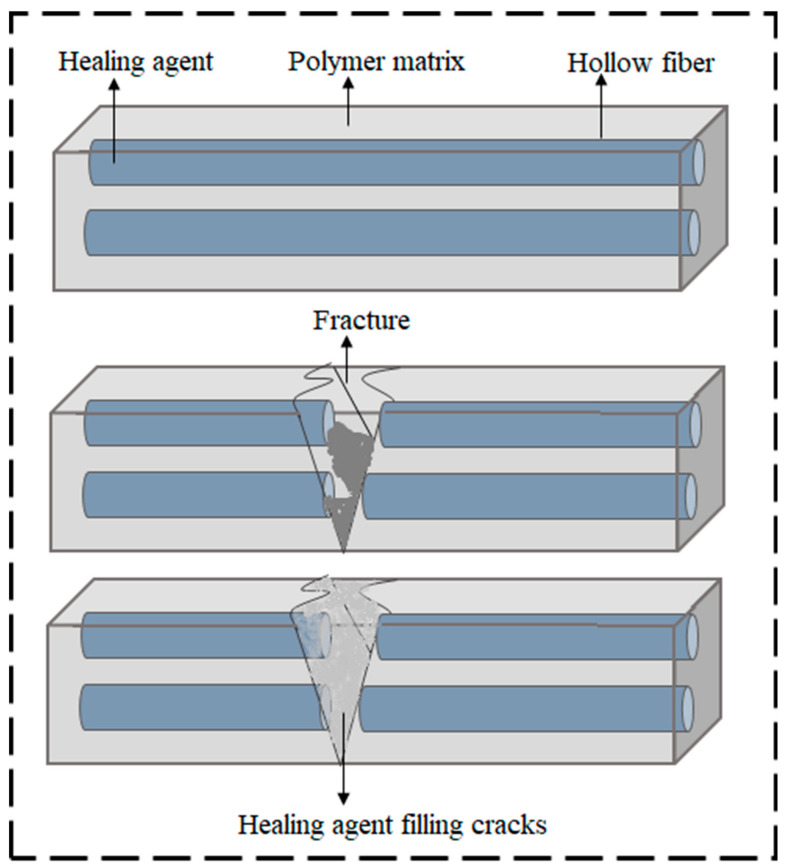
Representation of the self-healing concept using hollow-fiber embedment.

**Figure 4 ijms-23-00622-f004:**
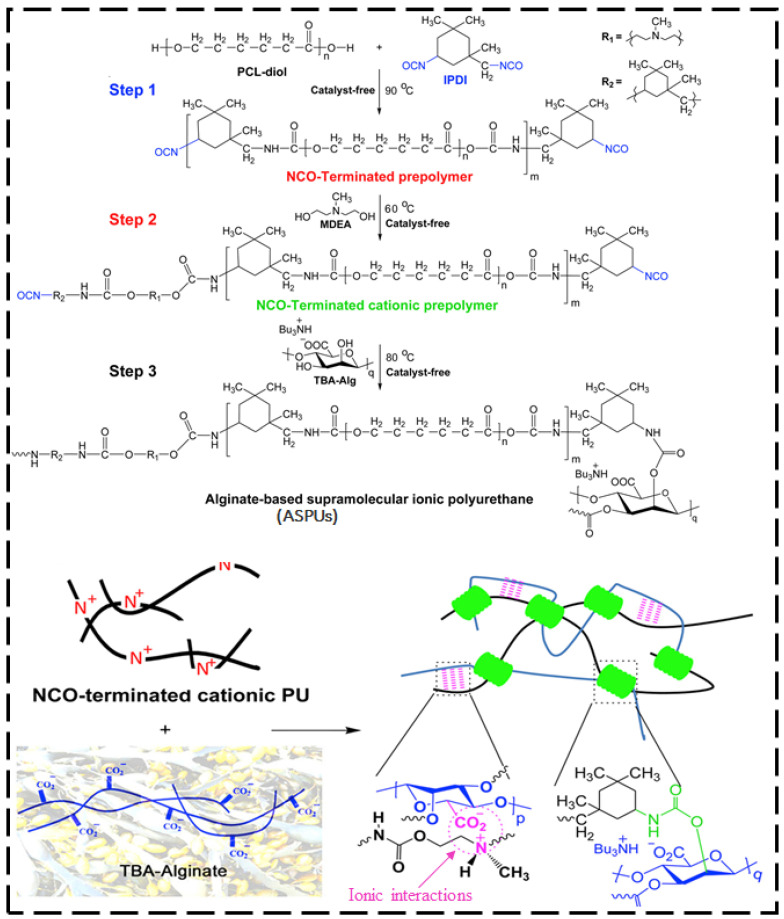
Alginate-based polyurethane self-healing elastomer that was designed by supramolecular ionic interactions. (Reproduced with permission [77] from Elsevier).

**Figure 6 ijms-23-00622-f006:**
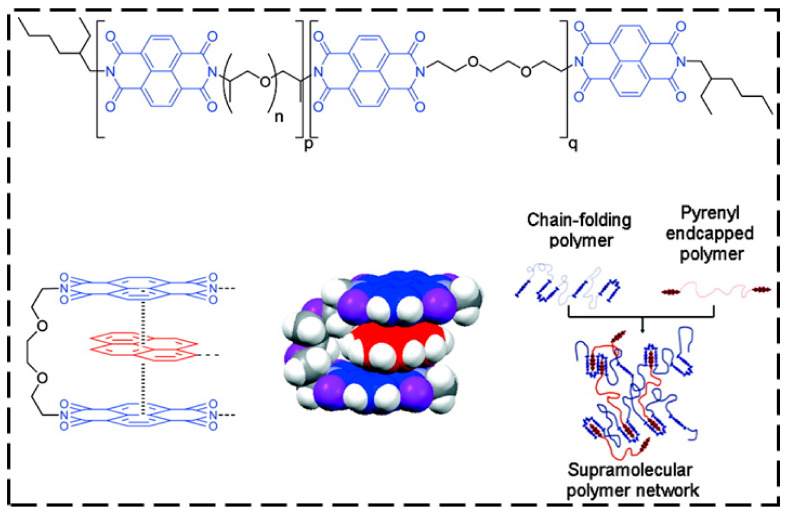
π–π stacking of π-electron-rich pyrenyl units with π-electron-deficient diimide groups. (Reproduced with permission [99] from American Chemical Society).

**Figure 8 ijms-23-00622-f008:**
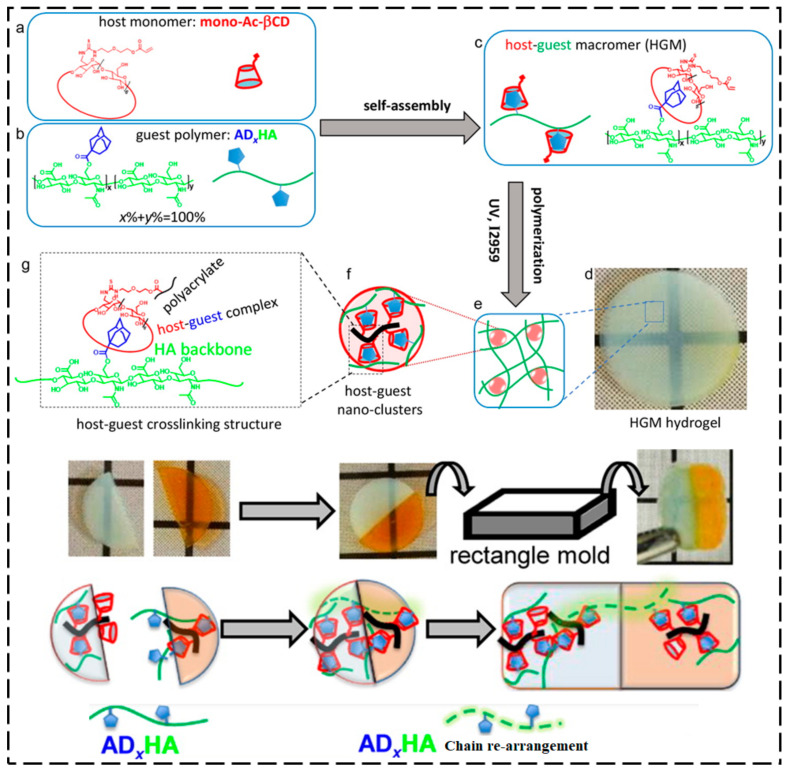
A self-healable supramolecular hydrogel that was based on adamantine-functionalized hyaluronic acid (ADxHA) as a guest polymer and monoacrylated β-cyclodextrin (mono-Ac-βCD) as a host monomer. (Reproduced with permission [123] from American Chemical Society).

**Figure 11 ijms-23-00622-f011:**
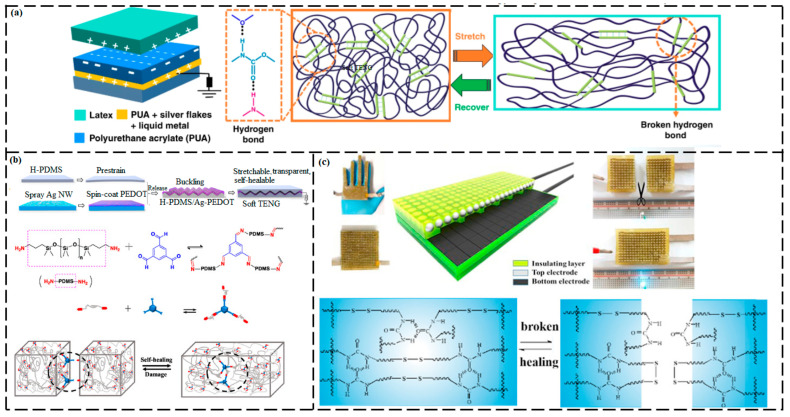
Self-healing triboelectric nanogenerators (TENGs). (**a**) Thermoplastic elastomer polyurethane acrylate-based self-healing TENG. (Reproduced with permission [182] from Springer Nature). (**b**) Poly(dimethylsiloxane)-based self-healing soft TENG. (Reproduced with permission [178] from American Chemical Society). (**c**) Magnetic-assisted self-healing TENG. (Reproduced with permission [183] from Elsevier).

**Figure 12 ijms-23-00622-f012:**
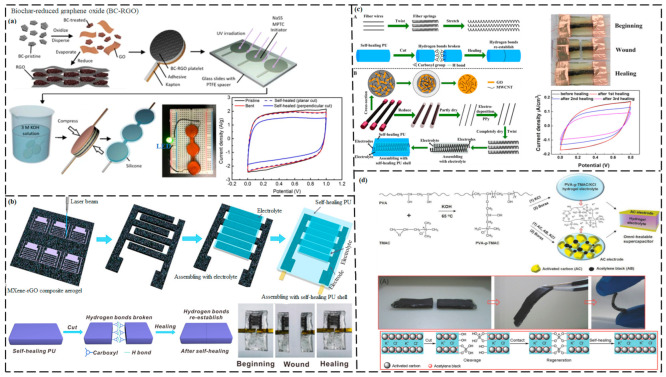
Self-healable SC. (**a**) Polyampholyte-based-SC with biochar–rGO electrodes. (Reproduced with permission [189] from Springer Nature). (**b**) Micro-SC with MXene–graphene aerogel electrode. (Reproduced with permission [190] from American Chemical Society). (**c**) SC with rGO-based fiber springs. (Reproduced with permission [191] from American Chemical Society). (**d**) Omni-healable SC. (Reproduced with permission [192] Wiley-VCH).

**Figure 14 ijms-23-00622-f014:**
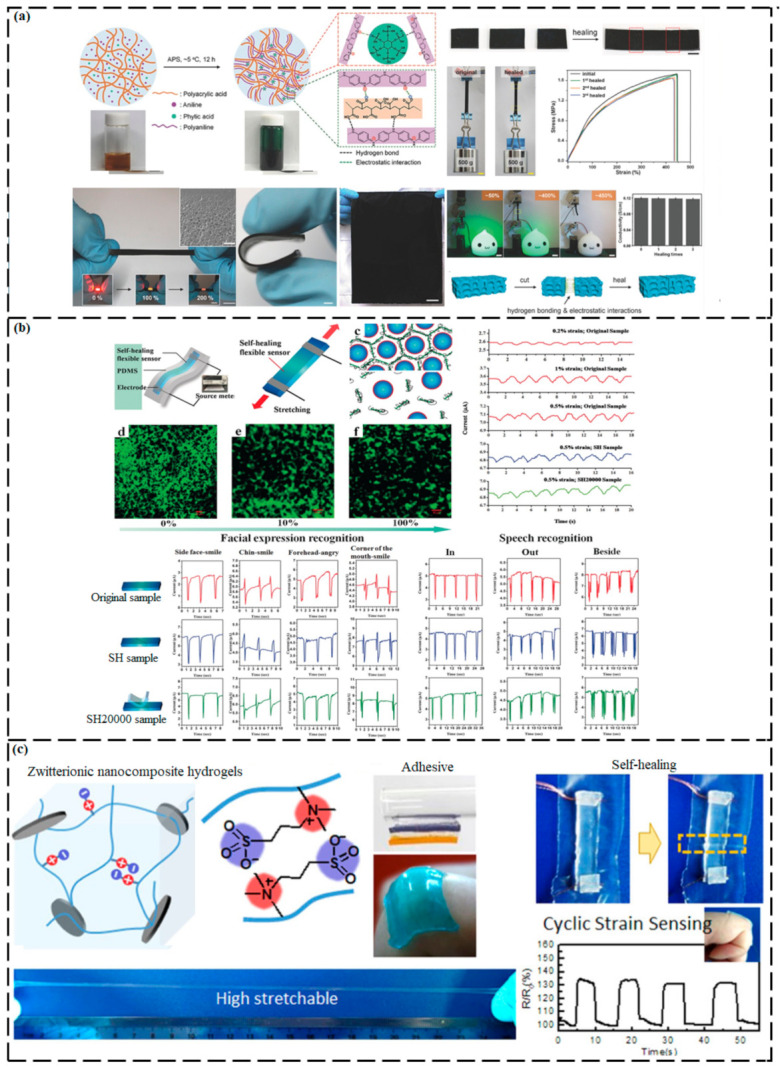
Self-healing sensors. (**a**) Self-healable ternary polymer composite for ultrasensitive strain and pressure sensing. (Reproduced with permission [213] Wiley-VCH). (**b**) Carboxyl cellulose/chitosan/epoxy natural rubber latex-based self-healing of sensors for human–machine interactions. (Reproduced with permission [81] from Wiley-VCH). (**c**) Zwitterionic nanocomposite hydrogels for strain sensing. (Reproduced with permission [214] from American Chemical Society).

## Data Availability

All data are contained within the article.

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
