# Peer review of "Self-Healing Materials for Electronics Applications"

_ijms, 2022, doi:10.3390/ijms23020622_

Round 1

Reviewer 1 Report

The authors review the researches about the self-healing materials for electronic applications in the present review manuscript. Some other reviews related to this issue can be found, this issue highlight the application of the self-healing applications. I would recommend this manuscript for publication after addressing some minor concerns:

  • Although authors report “This paper further surveys the numerous electronics applications of self-healing materials.”, I wouldn’t say such “numerous applications” can be observed (only 5 different applications).
  • Please check the typo at the end of page 6 (“inflitration; in addition,…”)
  • Section 2.1.3. In the way it is explained, this embedment appears quite similar to the microvascular embedment. Which is the difference? How “heal” the healing material? Is it related to the environmental humidity or any other reagent? If not, why doesnit it heal within the hollow-fiber? I would recommend to highlight this point.
  • Page 8, section 2.2.2.1 Ionic interaction. I guess that in this healing mechanism the melting enthalpy may play a critical role when selecting or designing a self-healing materials, but that issue seems to be not highlighted in this section (for example indicating which are the typical/minimum properties/requirements these materials should meet).
  • Pages 10, 13, 15, 17, 20. Please, correct the numbering of the titles (“Hydrogen bonding”,…).
  • Page 15, “Shao et al. [94] designed a nanocomposite hydrogel crosslinked by H-bonds and dual metal–carboxylate coordination bonds (be-tween iron ions (Fe3+) and carboxylic groups from poly(acrylic acid) and car-boxylated cellulose nanofibrils) (Figure 7a).” remove the extra parenthesis.
  • Figure 9a. Authors are required to reorganize Figure 9, since Figure 9a should be rotated to facilitate its readability.
  • There are so format problems, such as on top of the page 32 “2 of 54”.
  • The whole manuscript should have incorporaten the number of the lines. Apart from that, if “Sensors” will be the only application within 3 Bioelectronic devices, I would suggest to rename section 3.3 and to avoid the section 3.3.1.
  • Page 35 (but not only here). Authors should maintain the same font and size of the letter (see Funding).
  • Within the body of the text the authors refers to the last decade as the time-span of their review, but some of the works they include correspond to prior years (even further than 2000). I would recommend to check this point and update those outdated investigations.

Reviewer 2 Report

In this manuscript, the authors provided a comprehensive review of researches on Self-healing materials for electronics applications in various domains. The synthesis strategies and specific applications in energy-harvesting devices, energy storage devices, and bioelectronic devices are revealed. This manuscript could be helpful for the development of self-healing materials and devices for practical purposes.

I would like to recommend its publication in this journal after addressing the following recommendations:

1)         In some figures (for example 5, 9, 10, 11, 12, 13, and 14), the letters and/or formulas are too small to be clearly seen.

2)         For a better understanding of the self-healing mechanisms, the variety of self-healable natural and synthetic polymeric or co-polymeric materials, crosslinkers, spacers could be tabulated and systemized in a comprehensive table;

3)         In some sections additional information could be given. For example, in section 2.1. the size of the “microcapsules”, “fine hollow fibers” could be added.

4)         Please, highlight the input of your own research groups in the appropriate sections.

5)         Overall, a more critical view of the reviewed papers is needed.

6)         The future prospective of self-healing materials could be better emphasized.

Round 2

Reviewer 2 Report

The authors have carefully addressed all the review’s recommendations and the manuscript has been substantially improved.